# Attention Sparsity is Input-Stable: Training-Free Sparse Attention for Video Generation via Offline Sparsity Profiling and Online QK Co-Clustering

Jiayi Luo [1 2]  Jiayu Chen [3]  Jiankun Wang [4]  Cong Wang [5 2]  Hanxin Zhu [6]  Qingyun Sun [1]  Chen Gao [7 2]
Zhibo Chen [6 2 †]  Jianxin Li [1 †]

## Abstract

Diffusion Transformers (DiTs) achieve strong video generation quality but suffer from high inference cost due to dense 3D attention, leading to the development of sparse attention technologies to improve efficiency However, existing training-free sparse attention methods in video generation still face two unresolved limitations: *ignoring layer heterogeneity in attention pruning* and *ignoring query-key coupling in block partitioning*, which hinder a better quality-speedup trade-off. In this work, we uncover a critical insight that ***attention sparsity of each layer is its intrinsic property, with minor effects across different inputs***. Motivated by this, we propose **SVOO**, a training-free **S**parse attention framework for fast **V**ideo generation via **O**ffline layerwise sparsity profiling and **O**nline bidirectional co-clustering. Specifically, SVOO adopts a two-stage paradigm: (i) offline layer-wise sensitivity profiling to derive intrinsic per-layer pruning levels, and (ii) online block-wise sparse attention via a bidirectional co-clustering algorithm. Extensive experiments on seven widely used video generation models demonstrate that SVOO achieves a superior quality-speedup trade-off over state-of-the-art methods, delivering up to $1.93\times$ speedup while maintaining a PSNR of up to 29 dB on Wan2.1. Code is available at: https://github.com/Mutual-Luo/SVOO.

[1]SKLCCSE, School of Computer Science and Engineering, Beihang University [2]Zhongguancun Academy [3]School of Computer Science, Peking University [4]Beihang University [5]the State Key Laboratory of Multimodal Artificial Intelligence Systems, Institute of Automation, Chinese Academy of Sciences [6]School of Information Science and Technology, University of Science and Technology of China [7]BNRist, Tsinghua University. Correspondence to: Jianxin Li <lijx@buaa.edu.cn>, Zhibo Chen <chenzhibo@ustc.edu.cn>.

*Proceedings of the $43^{rd}$ International Conference on Machine Learning*, Seoul, South Korea. PMLR 306, 2026. Copyright 2026 by the author(s).

**Dense Attention | Latency: 418s**

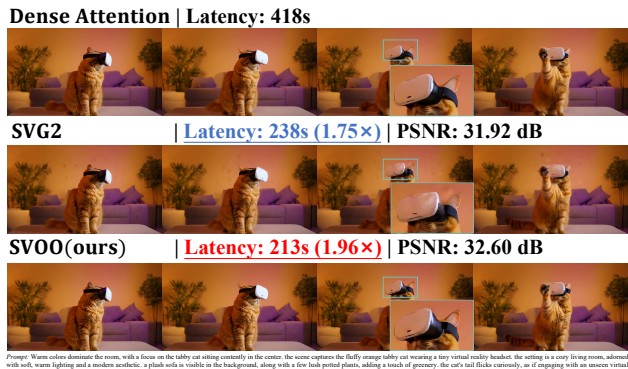

**SVG2** | Latency: 238s (1.75×) | **PSNR: 31.92 dB**

**SVOO(ours)** | Latency: 213s (1.96×) | **PSNR: 32.60 dB**

*Prompt: Warm colors dominate the room, with a focus on the tabby cat sitting contently in the center. The scene captures the fluffy orange tabby cat wearing a tiny virtual reality headset, the setting is a cozy living room, adorned with soft, warm lighting and a modern aesthetic. a plush sofa is visible in the background, along with a few lush potted plants, adding a touch of greenery. the cat's tail flicks curiously, as if engaging with an unseen virtual environment. its paws swipe at the air, indicating a playful and inquisitive nature, as it delves into the digital realm. the atmosphere is both whimsical and futuristic, highlighting the blend of analog and digital experiences.*

*Figure 1.* An example acceleration comparison on the Wan2.1-T2V-1.3B (Wan et al., 2025). All experiments are conducted on a single NVIDIA H200 GPU at a 720p resolution with 81 frames.

## 1. Introduction

Diffusion Transformers(DiTs) (Peebles & Xie, 2023) have already revolutionized the video generation, achieving high fidelity and temporal coherence. Despite their success (Wan et al., 2025; Yang et al., 2025b; Team et al., 2025; Wang et al., 2026), DiTs incur prohibitive computational overhead, primarily due to their dense 3D self-attention with quadratic complexity in spatial-temporal token count (Zhang et al., 2025c; Contributors, 2025; Team, 2024; Wang et al., 2025; Zou et al., 2025; Liu et al., 2025; Lin et al., 2024).

Recent studies mitigate the high computational cost of DiT attention by exploiting redundancy in attention mechanisms, motivated by empirical evidence that attention maps are highly sparse, with only a small fraction of attention weights being non-negligible (Zhang et al., 2025d; Sun et al., 2025). This observation has led to a series of sparse attention methods, which can be broadly divided into training-free approaches (Zhang et al., 2025a; Shen et al., 2025; Yang et al., 2025a; Xi et al., 2025; Xu et al., 2025; Li et al., 2025) and training-based approaches (Wu et al., 2025; Zhang et al., 2025f; Tan et al., 2025; Zhan et al., 2025), where training-free sparse attention reduces computation by directly leveraging sparsity without incurring additional training cost. In practice, training-free sparse attention is often implemented with a coarse-to-fine pipeline: tokens are partitioned into

blocks, block importance is estimated efficiently, and the dense attention is computed only for a subset of block pairs.

However, while these training-free sparse attention methods substantially reduce the inference cost of DiT-based video generation models, they still face two limitations:

- **L1: Ignore Layer Heterogeneity in Attention Pruning**: Most existing methods treat the multiple transformer layers as a homogeneous stack and apply uniform sparsity ratios across layers. In Sec. 3 and 4.1, we empirically and theoretically show that *attention sparsity is an intrinsic property of each layer, exhibiting pronounced variation across layers while remaining relatively stable within each layer across different inputs*. Such existing layer-agnostic designs overlook the distinct functional roles of different layers and their varying tolerance to attention pruning, leading to suboptimal sparsification decisions.
- **L2: Ignore Q-K Coupling in Block Partitioning**: Existing block-wise sparse attention methods partition queries and keys into blocks independently, despite the fact that salient attention patterns emerge from coupled Q-K interactions. In Sec. 3, our analysis indicates that *the optimal block partitioning of keys is query-dependent, and vice versa*. The existing Q-K decoupled blocking may misalign Q-K informative correspondences, leading to inferior sparsity patterns and reduced generation fidelity.

To address these challenges, we propose **SVOO**, a training-free **S**parse attention framework for fast **V**ideo generation via **O**ffline layer-wise sparsity profiling and **O**nline bidirectional co-clustering. Specifically, we first quantify the intrinsic pruning tolerance of each transformer layer via an offline calibration on a small set of random inputs, and derive a sparsity schedule that specifies the appropriate pruning ratio per layer. During inference, this schedule is applied to guide attention sparsification, delivering speedups with minor impact on quality. Next, to efficiently obtain a coupling-aware block partition without dense computation, we introduce a bidirectional co-clustering scheme that jointly groups queries and keys. Tokens are assigned according to their cross-attention affinity to opposite-side centroids and iteratively refined, producing well-aligned Q-K blocks with negligible overhead. We evaluate SVOO on 7 widely used models, including Wan2.1-T2V-1.3B, Wan2.1-T2V-14B, Wan2.1-I2V-14B, Wan2.2-T2V-A14B, Wan2.2-I2V-A14B, HunyuanVideo-T2V, and HunyuanVideo-I2V. SVOO consistently achieves a better trade-off between quality and efficiency than the state-of-the-art training-free sparse attention methods. Our contributions are summarized as follows:

- We conduct an in-depth analysis of existing training-free sparse attention methods and reveal two unresolved limitations: ignoring layer heterogeneity in attention pruning and overlooking Q-K coupling in block partitioning.
- We propose SVOO, a novel training-free sparse attention

framework tailored for fast video generation via offline layer-wise sparsity profiling and online bidirectional co-clustering to solve the aforementioned two limitations.
- Extensive experiments across 7 widely used video generation models demonstrate that SVOO offers a better trade-off between generation quality and inference speedup.

## 2. Related Work

Existing sparse attention methods for fast video generation can be broadly categorized into training-based sparse attention and training-free sparse attention. In this work, our method focuses on training-free sparse attention methods.

### 2.1. Training-based Sparse Attention

A recent line of work focuses on trainable sparse attention, where sparsity patterns are learned during training. VMoBA (Wu et al., 2025) accelerates video DiTs by employing a layer-wise recurrent 1D-2D-3D block partitioning scheme and a global threshold-based selection strategy. VSA (Zhang et al., 2025e) learns an end-to-end block-sparse attention using a coarse-to-fine tile selection scheme with annealed dense-to-sparse training. DSV (Tan et al., 2025) trains per-module predictors to approximate attention scores and pre-select critical key-value pairs, enabling fused sparse attention and sparsity-aware parallelism for faster training. BSA (Zhan et al., 2025) jointly sparsifies queries and key-value blocks via semantic query selection and dynamic KV thresholding under an annealed sparsity schedule to accelerate training and inference. While their promising speedups, training-based methods introduce extra training overhead.

### 2.2. Traning-free Sparse Attenion

Training-free methods reduce inference computation by directly exploiting attention sparsity without introducing additional training cost. STA (Zhang et al., 2025f) introduces a hardware-friendly sliding-tile attention mechanism that replaces global 3D attention with window-based blocks. SVG (Xi et al., 2025) classifies attention heads into spatial or temporal groups using an efficient profiling strategy. Radial (Li et al., 2025) applies a multi-band mask with radially shrinking attention windows and sampling frequencies over time. RainFusion (Chen et al., 2025) identifies a small set of important key-value tokens, often corresponding to motion regions or high-frequency textures, and extends the patterns in SVG. AdaSpa (Xia et al., 2025) performs an online search for effective sparse patterns by exploiting the cross-step invariance of attention. SpargeAttn (Zhang et al., 2025b) and DraftAttention (Shen et al., 2025) estimate block importance using aggregated token activations and skip low-score attention blocks to reduce computation. XAttention (Xu et al., 2025) identifies critical attention blocks using an efficient antidiagonal-sum proxy for block importance, combined

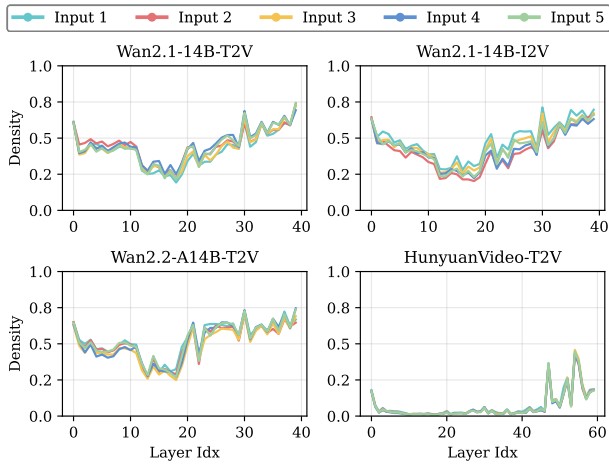

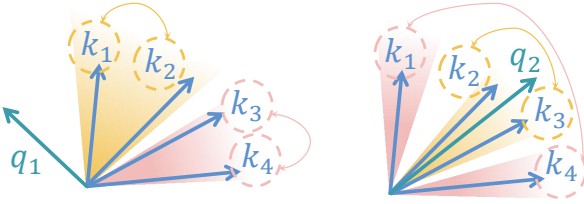

**The optimal block partitioning of Keys is Query-dependent.**

Given $q_1$, the optimal Key blocks partition are $[k_1, k_2], [k_3, k_4]$

Given $q_2$, the optimal Key blocks partition are $[k_2, k_3], [k_1, k_4]$

*Figure 3.* Illustration of query-key coupling in block partitioning, which is important for block-wise sparse attention. The example shows that **the optimal partitioning of Keys is Query-dependent**, as different Queries induce different optimal groupings of Keys.

*Figure 2.* Layer-wise attention sparsity across different models. The figure shows that attention density varies substantially across layers (*layer-wise heterogeneity*), while **remaining highly stable for each layer across different inputs (*layer-wise stability*)**.

with a dynamic programming-based thresholding strategy. SVG2 (Yang et al., 2025a) improves block-wise sparse attention by grouping semantically similar tokens via k-means clustering into contiguous memory layouts. However, existing training-free methods still overlook layer heterogeneity in attention pruning and Q-K coupling in block partitioning, resulting in a suboptimal quality-efficiency trade-off.

## 3. Motivation and Analysis

Here, we conduct the in-depth analysis of our motivations introduced in Sec. 1: (i) layer heterogeneity in attention pruning and (ii) Q-K coupling in block partitioning.

### 3.1. Layer Heterogeneity in Attention Pruning

To examine layer-wise heterogeneity in pruning tolerance, we conduct an empirical study on four representative video generation models: Wan2.1-14B-T2V, Wan2.1-14B-I2V, Wan2.2-A14B-T2V (Wan et al., 2025), and HunyuanVideo-T2V (Team et al., 2025). For each layer, we measure the attention density as the minimum fraction of attention entries needed to cover 80% of the cumulative attention mass, i.e., the number of selected positions divided by the total number of entries in the attention map (Yang et al., 2025a). For the results in Figure 2, we randomly sample 5 prompts from VBench (Zheng et al., 2025; Huang et al., 2024) per model. Additional details are provided in the Appendix B.

The results in Figure 2 lead to two key observations:

- *Layer-wise Heterogeneity*: Attention density varies substantially across layers, indicating that **different layers contribute unevenly to the overall attention computation**. This variation implies that transformer layers exhibit

markedly different tolerance levels to attention pruning when accelerating video generation, and thus applying a uniform sparsity ratio across all layers may lead to inefficient or overly aggressive pruning in certain transformer layers.

- *Layer-wise Stability*: For a certain layer, the measured **attention sparsity remains highly consistent across different inputs**, suggesting that the attention pruning tolerance of each layer is largely invariant to input content. This stability indicates that each layer's sparsity pattern is governed more by its architectural role and learned parameters than by the specific input characteristics.

In Sec. 4.1, we further provide a theoretical analysis of this phenomenon. Together, these observations imply that each DiT layer exhibits a distinct yet relatively stable sparsity characteristic across diverse inputs. *This insight motivates sparse attention designs to account for layer-wise heterogeneity rather than applying uniform pruning across layers.*

### 3.2. Q-K coupling in Block Partitioning

Existing block-wise sparse attention methods typically follow a two-stage pipeline: they first partition queries and keys into blocks to obtain a coarse block-level attention estimate, and then compute dense attention only for a small set of selected block pairs. The quality of block partitioning is therefore crucial, as it determines whether coarse estimates can reliably reflect true attention strength and whether the selected blocks cover the dominant semantic dependencies. Ideally, queries within the same query block should exhibit similar attention preferences, and keys within the same key block should be attended by similar queries, so that attention mass concentrates on a few well-aligned block pairs.

To build intuition, consider a simple example. For a fixed query $\mathbf{q}$, two keys $\mathbf{k}_1$ and $\mathbf{k}_2$ should be grouped into the same block if they yield similar attention logits, i.e.,

$$\mathbf{q}^\top \mathbf{k}_1 \approx \mathbf{q}^\top \mathbf{k}_2 \iff \mathbf{q}^\top (\mathbf{k}_1 - \mathbf{k}_2) \approx 0. \quad (1)$$

Here $\mathbf{q}^\top (\mathbf{k}_1 - \mathbf{k}_2) \approx 0$ implies that placing $\mathbf{k}_1$ and $\mathbf{k}_2$ into

the same key block depends on whether $(\mathbf{k}_1 - \mathbf{k}_2)$ has a small inner product with $\mathbf{q}$, making key-block partitioning inherently query-dependent. As illustrated in Figure 3, the optimal key partition can vary across queries: for given query $\mathbf{q}_1$, a suitable block partition of keys is $[\mathbf{k}_1, \mathbf{k}_2]$ and $[\mathbf{k}_3, \mathbf{k}_4]$, whereas for given query $\mathbf{q}_2$, the optimal block partition of keys may shift to $[\mathbf{k}_1, \mathbf{k}_4]$ and $[\mathbf{k}_2, \mathbf{k}_3]$. Therefore, independently partitioning queries and keys can introduce structural mismatch and fragment high-mass attention regions, motivating a joint, coupling-aware block partitioning.

# 4. Our Proposed SVOO

In this section, we elaborate on our proposed **SVOO**, a novel training-free **S**parse attention framework for fast **V**ideo generation via **O**ffline layer-wise sparsity profiling and **O**nline bidirectional co-clustering. Specifically, SVOO adopts a two-stage paradigm: it first profiles the intrinsic attention sparsity of each transformer layer offline to derive a layer-specific sparsity schedule, and then performs online bidirectional co-clustering during inference to construct coupling-aware query-key block partitions and identify salient attention blocks under this schedule efficiently. An overview of our proposed SVOO framework is shown in Figure 4.

## 4.1. Offline Layer-Wise Sparsity Profiling

Based on the analysis in Sec. 1, we empirically identify two key properties of DiT layers: *layer-wise heterogeneity* and *layer-wise stability*. Layer-wise heterogeneity refers to the substantial variation in attention sparsity across different layers, while layer-wise stability indicates that the sparsity pattern of a given layer remains highly consistent across different inputs, reflecting an intrinsic property of that layer.

To theoretically analyze these phenomena, we focus on the statistics of the pre-softmax attention logits. Let $\mathbf{X} \in \mathbb{R}^{n \times d}$ denote the input representation to a transformer layer with $n$ tokens. For a query token $i$, the pre-softmax logits are $\mathbf{a}_i = \mathrm{softmax}\big(\mathbf{z}_i(\mathbf{X})\big)$, where $\mathbf{z}_i(\mathbf{X}) = \frac{(\mathbf{x}_i \mathbf{W}_Q)(\mathbf{X}\mathbf{W}_K)^\top}{\sqrt{d'}}$ with $\mathbf{x}_i$ is the $i$-th row of $\mathbf{X}$, $d'$ is the attention head dimension and $\mathbf{W}_Q, \mathbf{W}_K$ are the query and key projection matrices. Since softmax is applied row-wise, the concentration of $\mathbf{a}_i$ is largely governed by the dispersion of $\mathbf{z}_i(\mathbf{X})$: a larger $\mathrm{Var}(\mathbf{z}_i)$ typically yields a more peaked $\mathbf{a}_i$ (i.e., higher sparsity), while a smaller $\mathrm{Var}(\mathbf{z}_i)$ produces a flatter, more uniform distribution. Accordingly, we adopt the average row-wise variance of the pre-softmax logits as a proxy for attention sparsity, which we denote by $V(\mathbf{X})$ and define as:

$$V(\mathbf{X}) \triangleq \frac{1}{n} \sum_{i=1}^{n} \mathrm{Var}\big(\mathbf{z}_i(\mathbf{X})\big). \tag{2}$$

To analyze layer-wise heterogeneity and stability theoretically, we first introduce the following Assumption 4.1:

**Assumption 4.1** (Bounded Token Representations). Consider a transformer layer and let its input be $\mathbf{X} \in \mathbb{R}^{n \times d}$, whose rows are token representations $\mathbf{x}_i \in \mathbb{R}^{1 \times d}$. Assume $\{\mathbf{x}_i\}_{i=1}^{n}$ are from a distribution on $\mathbb{R}^d$ with population mean $\boldsymbol{\mu}_\star \triangleq \mathbb{E}[\mathbf{x}]$ and covariance $\boldsymbol{\Sigma}_\star \triangleq \mathbb{E}\big[(\mathbf{x} - \boldsymbol{\mu}_\star)^\top (\mathbf{x} - \boldsymbol{\mu}_\star)\big]$. We assume that there exists a constant $R > 0$ such that:

$$\|\mathbf{x}\|_2 \leq R. \tag{3}$$

Assumption 4.1 assumes that, within a well-trained layer, token representations are contained in an $R$-ball around the population mean. *Such a boundedness assumption is reasonable in practice*, as normalization layers (e.g., Layer-Norm/RMSNorm) and residual connections stabilize activation magnitudes in existing DiT-based video generation models. Then, we propose the following Theorem 4.2:

**Theorem 4.2** (Layer-wise Sparsity Stability). *Consider a well-trained transformer layer and denote by $V(\mathbf{X})$ the average row-wise variance of the pre-softmax attention logits produced by this layer for input $\mathbf{X} \in \mathbb{R}^{n \times d}$. Under Assumption 4.1, for any two independent inputs $\mathbf{X}, \hat{\mathbf{X}} \in \mathbb{R}^{n \times d}$ of equal token length, it holds with probability at least $1 - \delta$:*

$$\begin{aligned} &\big|V(\mathbf{X}) - V(\hat{\mathbf{X}})\big| \\ &\leq \frac{d\,\|\mathbf{M}\|_2^2}{d'}\, C\, R^4 \left( \sqrt{\frac{\log(d/\delta)}{n}} + \frac{\log(d/\delta)}{n} \right), \end{aligned} \tag{4}$$

*where $C > 0$ is an absolute constant, $\mathbf{M} \triangleq \mathbf{W}_Q \mathbf{W}_K^\top$ with $\mathbf{W}_Q, \mathbf{W}_K \in \mathbb{R}^{d \times d'}$ are the query and key projection matrices of this layer, and $d'$ is the attention head dimension.*

Theorem 4.2 provides an upper bound on the discrepancy between the pre-softmax attention logit variances induced by two different inputs, with its detailed proof can be found in Appendix A. Importantly, the bound depends on $\mathbf{M}$, implying that this discrepancy is influenced by layer-specific parameters, yielding *layer heterogeneity*. Meanwhile, in video generation scenarios, the token length $n$ is typically very large and satisfies $R \ll n$, which makes the RHS of Eq. (4) small. As a result, within each layer, the logit variance is largely input-invariant, yielding *layer stability*.

Motivated by the aforementioned layer heterogeneity and stability, we introduce an offline profiling module to derive a reliable layer-head sparsity schedule. The schedule aims to avoid overly aggressive pruning in sensitive layers while preventing insufficient sparsification in redundant ones. We first construct a small calibration set $\mathcal{D} = \{x^k\}_{k=1}^{m}$ by randomly sampling $m$ inputs. For each layer $\ell$, head $h$, and calibration input $x^k$, we compute an attention density $d_{\ell,h}^k \in (0, 1]$, defined as the minimum fraction of attention entries required to cover a proportion $\tau$ of the cumulative attention mass where we set $\tau = 0.95$. Specifically, let $\mathbf{A}_{\ell,h}^k \in \mathbb{R}^{n \times n}$ denote the post-softmax attention matrix of layer $\ell$ and head

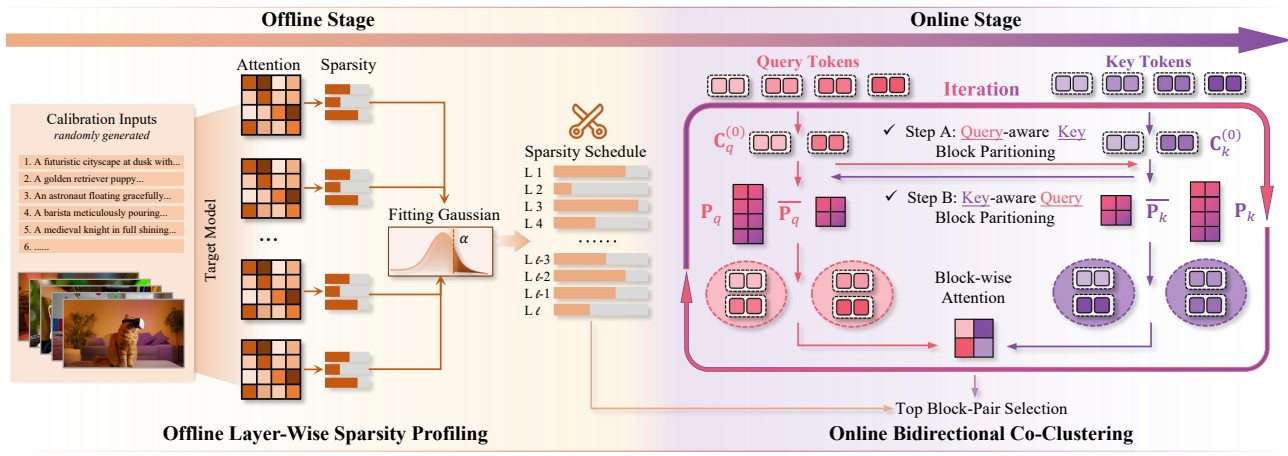

*Figure 4.* The framework of SVOO consists of two stages for accelerating video generation. Offline stage (left): we profile the intrinsic attention sparsity of each transformer layer and derive a layer-wise sparsity schedule. Online stage (right): we perform bidirectional co-clustering to partition queries and keys into coupled blocks, and then select salient block pairs according to the offline schedule.

$h$ under input $x^k$. For each row $i$, we sort $\{\mathbf{A}_{\ell,h}^k(i,j)\}_{j=1}^n$ in descending order and define $\mathcal{S}_{\ell,h}^{(j)}(i) \subseteq \{1, \ldots, n\}$ as the minimal prefix satisfying $\sum_{k \in \mathcal{S}_{\ell,h}^{(j)}(i)} A_{\ell,h}^{(j)}(i,k) \geq \tau$. The attention density $d_{\ell,h}^{(j)}$ is then computed as follows:

$$d_{\ell,h}^{(j)} = \frac{1}{n} \sum_{i=1}^{n} \frac{|\mathcal{S}_{\ell,h}^{(j)}(i)|}{n}. \tag{5}$$

We then fit a univariate Gaussian $d_{\ell,h}^{(j)} \sim \mathcal{N}(\mu_{\ell,h}, \sigma_{\ell,h}^2)$ to $\{d_{\ell,h}^{(j)}\}_{j=1}^m$, and take the upper $\alpha$-quantile as a conservative estimate $\hat{d}_{\ell,h} = \mu_{\ell,h} + z_\alpha \sigma_{\ell,h}$, with $\alpha = 0.95$. Finally, we derive the sparsity schedule $s_{\ell,h}$ as follows:

$$s_{\ell,h} = 1 - \hat{d}_{\ell,h}, \tag{6}$$

which is used to guide the subsequent online attention sparsification. Notably, the calibration set $\mathcal{D}$ can be arbitrary reasonable inputs due to the layer-wise stability of DiTs.

### 4.2. Online Bidirectional Co-Clustering

As analyzed in Section 3, existing sparse attention methods partition queries and keys independently, ignoring the intrinsic Q-K coupling that shapes the resulting attention map. Such Q-K decoupled blocking often yields suboptimal partitions, as tokens grouped within the same block may exhibit large cross-attention disparities, making subsequent block selection less reliable. Ideally, queries within a block should share similar attention preferences, while keys within a block should exhibit similar relevance across queries.

To this end, we design a novel bidirectional co-clustering algorithm that explicitly accounts for Q-K coupling during block partitioning, with its details provided in Algorithm 1.

Specifically, the proposed bidirectional co-clustering algorithm alternates between query-aware key partitioning and key-aware query partitioning to jointly align query and key blocks. Given query tokens $\mathcal{Q} = \{\mathbf{q}_i\}_{i=1}^N$ and key tokens $\mathcal{K} = \{\mathbf{k}_j\}_{j=1}^N$, we first initialize the query and key block centroids $\mathbf{C}_q^{(0)}$ and $\mathbf{C}_k^{(0)}$ by randomly sampling anchor tokens. At each iteration, we first perform *query-aware key-side clustering*. For each key token, we compute its affinity vector $\mathbf{P}_k$ with respect to the current query centroids $\mathbf{C}_q^{(i-1)}$, which characterizes how the key is attended by different query blocks. Keys are then assigned to the nearest key centroid by comparing these affinity patterns, yielding updated key block assignments $\mathcal{L}_k$ and centroids $\mathbf{C}_k^{(i)}$. This step groups together keys that exhibit similar relevance across queries. Next, we perform *key-aware query-side clustering* in a symmetric manner. By iteratively alternating between these two steps, the algorithm jointly refines query and key partitions in a coupling-aware fashion. As a result, queries within the same block share similar attention preferences, while keys within the same block exhibit similar relevance across queries, producing well-aligned query-key blocks.

**Top Block-Pair Selection**. We perform the block-wise selection via coarse-grained estimation $\bar{\mathbf{A}} = \mathbf{C}_q \mathbf{C}_k^\top$. To balance the recall $\tau$ and offline budget $s_{l,h}$, the selection ratio $\rho_{l,h}$ is determined by a threshold-dependent strategy:

$$\rho_{l,h} = \begin{cases} \min(\text{Recall}(\bar{\mathbf{A}}, \tau), s_{l,h}), & \text{if } s_{l,h} > \theta \\ \max(\text{Recall}(\bar{\mathbf{A}}, \tau), s_{l,h}), & \text{if } s_{l,h} \leq \theta \end{cases} \tag{7}$$

Dense attention is computed only over the top $\rho_{l,h} K_k$ blocks

**Clustering Reuse.** While bidirectional co-clustering incurs extra computation, we find that our resulting block partitions are highly stable across diffusion steps. We therefore reuse the clustering results and recompute them every $N$ steps.

**Algorithm 1:** Bidirectional Co-Clustering Algorithm for the Query-Key Coupled Block Partitioning.

---

**Input:** Query tokens $\mathcal{Q} = \{\mathbf{q}_i\}_{i=1}^{N}$, Key tokens
$\quad\quad \mathcal{K} = \{\mathbf{k}_j\}_{j=1}^{N}$; Target number of blocks
$\quad\quad K_q, K_k$; Max iterations $I_{\max}$.
**Output:** Query block assignments $\mathcal{L}_q$; Key block
$\quad\quad$ assignments $\mathcal{L}_k$; Block centroids $\mathbf{C}_q, \mathbf{C}_k$.

1 $\mathbf{C}_q^{(0)} \leftarrow \text{Sample}(\mathcal{Q}, K_q); \quad \mathbf{C}_k^{(0)} \leftarrow \text{Sample}(\mathcal{K}, K_k)$ ;
2 **for** $i = 1$ **to** $I_{max}$ **do**
$\quad\quad$ ▶ **Step A: Query-aware Key-side Block Partitioning**
3 $\quad\quad \mathbf{P}_k \leftarrow \mathcal{K}(\mathbf{C}_k^{(i-1)})^\top; \bar{\mathbf{P}}_k \leftarrow \mathbf{C}_k^{(i-1)}(\mathbf{C}_q^{(i-1)})^\top$ ;
$\quad\quad$ // *Affinity to query anchors*
4 $\quad\quad \mathbf{P}_k \leftarrow \text{Norm}(\mathbf{P}_k); \bar{\mathbf{P}}_k \leftarrow \text{Norm}(\bar{\mathbf{P}}_k)$
5 $\quad\quad \mathcal{L}_k \leftarrow \arg\min_{j \in \{1,\dots,K_k\}} \|\mathbf{P}_k - \bar{\mathbf{P}}_k[j]\|_2$ ;
$\quad\quad$ // *Assign keys to blocks*
6 $\quad\quad \mathbf{C}_k^{(i)} \leftarrow \text{Mean}(\mathcal{K} \text{ via } \mathcal{L}_k)$ ;
$\quad\quad$ ▶ **Step B: Key-aware Query-side Block Partitioning**
7 $\quad\quad \mathbf{P}_q \leftarrow \mathcal{Q}(\mathbf{C}_k^{(i)})^\top; \bar{\mathbf{P}}_q \leftarrow \mathbf{C}_q^{(i-1)}(\mathbf{C}_k^{(i)})^\top$ ;
$\quad\quad$ // *Affinity to key anchors*
8 $\quad\quad \mathbf{P}_q \leftarrow \text{Norm}(\mathbf{P}_q); \bar{\mathbf{P}}_q \leftarrow \text{Norm}(\bar{\mathbf{P}}_q)$
9 $\quad\quad \mathcal{L}_q \leftarrow \arg\min_{j \in \{1,\dots,K_q\}} \|\mathbf{P}_q - \bar{\mathbf{P}}_q[j]\|_2$ ;
$\quad\quad$ // *Assign queries to blocks*
10 $\quad\quad \mathbf{C}_q^{(i)} \leftarrow \text{Mean}(\mathcal{Q} \text{ via } \mathcal{L}_q)$ ;
11 **end**
12 **return** $\mathcal{L}_q, \mathbf{C}_q, \mathcal{L}_k, \mathbf{C}_k$

---

**Kernel Customization.** We implement the bidirectional co-clustering algorithm using Triton, and adopt dynamic block-size FlashInfer kernels from prior work (Yang et al., 2025a; Ye et al., 2025) for block-wise sparse attention computation.

**Difference from SVG2.** SVG2 (Yang et al., 2025a) independently partitions queries and keys into blocks using K-means, ignoring that the optimal block partitioning of queries varies with keys, and vice versa. In contrast, we account for Q-K coupling by jointly partitioning queries and keys via novel bidirectional co-clustering, yielding better-aligned blocks with similar attention preferences within each block.

## 5. Experiment

### 5.1. Setup

**Models.** We evaluate SVOO on 7 widely used video generation models, covering both text-to-video and image-to-video tasks. Specifically, the text-to-video models include Wan2.1-1.3B-T2V, Wan2.1-14B-T2V, Wan2.2-A14B-T2V (Wan et al., 2025), and HunyuanVideo-T2V (Team et al., 2025), while image-to-video models include Wan2.1-14B-I2V, Wan2.2-A14B-I2V, and HunyuanVideo-I2V.

**Metrics.** We evaluate both the quality and efficiency of SVOO and the baselines. For quality assessment, we use Peak Signal-to-Noise Ratio (PSNR), Learned Perceptual Image Patch Similarity (LPIPS) (Zhang et al., 2018), and Structural Similarity Index Measure (SSIM) (Wang et al., 2004) to measure the similarity between videos generated with sparse and dense attention. In addition, we adopt the VBench score (Huang et al., 2024) to evaluate video generation quality, reporting metrics on image quality, aesthetic quality, subject consistency, and background consistency. For efficient assessment, we report the inference latency and the overall speedup achieved under the same settings.

**Datasets.** For text-to-video generation task, we follow the Penguin Benchmark with prompt optimization provided by the VBench team (Huang et al., 2024), while for image-to-video generation task, we use the prompt-image pairs from VBench++ (Zheng et al., 2025) with a 16:9 aspect ratio.

**Baselines.** We compare SVOO with state-of-the-art training-free sparse attention methods tailored for accelerated video generation, including SpargeAttention (Zhang et al., 2025a), SVG (Xi et al., 2025), SVG2 (Yang et al., 2025a), and Radial (Yang et al., 2025a). We use official configurations from their open-sourced repositories, except for unified warm-up.

**Implementations.** We set the number of query and key blocks to $K_q = 256$ and $K_k = 1024$, respectively. The bidirectional co-clustering algorithm is run with only two iterations per clustering and recomputed every 20 diffusion steps. The threshold is set to $\theta = 0.1$. For all models, we apply a layer warm-up of one layer, where the first layer always uses the dense attention. In addition, Wan-series models use 20% dense attention warm-up diffusion steps, while HunyuanVideo-series models use a 10% warm-up. For our proposed SVOO, SpargeAttention, SVG, and SVG2, videos are generated at a standard 720p resolution ($720 \times 1280$). Specifically, Wan-series models generate 81-frame videos, while HunyuanVideo-series models generate 129-frame videos. For Radial, due to constraints imposed by its acceleration strategy, we follow its specification and use a resolution of $704 \times 1280$, with 85 frames for Wan-series models and 133 frames for HunyuanVideo-series models. All experiments are conducted on the NVIDIA H200 GPU.

### 5.2. Quality and Efficiency Evaluation

We evaluate the quality and efficiency of SVOO and the baselines on both text-to-video and image-to-video generation tasks, with results reported in Tables 1 and 2, respectively. Overall, SVOO consistently achieves the highest speedup among all methods while maintaining better generation quality in most settings. For Wan2.1-1.3B-T2V, SVOO attains a $1.93\times$ speedup, outperforming the runner-up SVG2 by $0.20\times$, and improves image quality and aesthetic quality by 4.74% and 4.30%, respectively. Moreover, we provide qual-

*Table 1.* Quality and efficiency benchmarking results of our proposed SVOO and baselines on Text-to-Video Task.

| Model | Config | Baseline | Quality | | | | | | | Efficiency | |
|---|---|---|---|---|---|---|---|---|---|---|---|
| | | | PSNR↑ | SSIM↑ | LPIPS↓ | ImageQual↑ | AesQual↑ | SubConsist↑ | BackConsist↑ | Latency | Speedup |
| Wan2.1 | 1.3B-720P-T2V | Origin | - | - | - | 66.58% | 64.47% | 96.74% | 97.30% | 417s | 1.00× |
| | | SpargeAttn | 25.137 | 0.801 | 0.234 | 63.08% | 62.75% | 95.79% | 97.09% | 288s | 1.45× |
| | | SVG1 | 25.712 | 0.811 | 0.230 | 63.26% | 61.14% | 94.27% | 91.16% | 266s | 1.56× |
| | | SVG2 | 29.268 | 0.886 | 0.127 | 61.83% | 60.15% | 95.88% | 96.47% | 241s | 1.73× |
| | | Radial | 26.305 | 0.829 | 0.182 | 63.67% | 62.76% | 96.56% | 97.07% | 257s | 1.62× |
| | | **SVOO** | **29.986** | **0.898** | **0.125** | **66.57%** | **64.45%** | **96.62%** | **97.19%** | **216s** | **1.93×** |
| | 14B-720P-T2V | Origin | - | - | - | 69.14% | 61.27% | 97.64% | 97.70% | 1982s | 1.00× |
| | | SpargeAttn | 23.837 | 0.751 | 0.210 | 64.48% | 57.91% | 96.93% | 97.25% | 1394s | 1.42× |
| | | SVG1 | 23.957 | 0.804 | 0.194 | 67.80% | 58.95% | 97.09% | 97.12% | 1239s | 1.60× |
| | | SVG2 | 27.342 | 0.892 | **0.111** | 68.29% | 59.06% | 97.24% | 97.07% | 1261s | 1.57× |
| | | Radial | 23.358 | 0.798 | 0.206 | 67.66% | 60.83% | 97.55% | 97.49% | 1297s | 1.53× |
| | | **SVOO** | **27.786** | **0.893** | **0.111** | **68.92%** | **61.01%** | **97.67%** | **97.69%** | **1203s** | **1.64×** |
| Wan2.2 | 14B-720P-T2V | Origin | - | - | - | 72.62% | 65.21% | 97.24% | 97.41% | 1608s | 1.00× |
| | | SpargeAttn | 19.638 | 0.697 | 0.288 | 70.77% | 63.29% | 96.18% | 96.68% | 1116s | 1.44× |
| | | SVG1 | 21.293 | 0.798 | 0.210 | 71.84% | 65.13% | 96.16% | 96.67% | 1049s | 1.53× |
| | | SVG2 | 24.477 | 0.856 | 0.142 | 71.51% | 62.17% | 96.22% | 96.45% | 1061s | 1.52× |
| | | Radial | 20.452 | 0.704 | 0.270 | 71.47% | 64.76% | 95.78% | 96.22% | 1164s | 1.38× |
| | | **SVOO** | **24.846** | **0.860** | **0.144** | **72.92%** | **65.16%** | **96.72%** | **97.01%** | **984s** | **1.63×** |
| Hunyuan | 13B-720P-T2V | Origin | - | - | - | 67.56% | 57.28% | 97.67% | 97.76% | 1783s | 1.00× |
| | | SpargeAttn | 22.394 | 0.770 | 0.236 | 66.95% | 55.22% | 96.49% | 96.64% | 1294s | 1.38× |
| | | SVG1 | 21.979 | 0.752 | 0.259 | 67.04% | 55.56% | 96.47% | 96.51% | 897s | 1.99× |
| | | SVG2 | **25.218** | 0.841 | **0.205** | 66.90% | 55.31% | 96.78% | 96.49% | 909s | 1.96× |
| | | Radial | 24.319 | 0.805 | 0.219 | 66.33% | **56.74%** | 97.04% | 96.72% | 916s | 1.94× |
| | | **SVOO** | 24.879 | **0.843** | 0.224 | **67.93%** | 55.80% | **97.99%** | **97.50%** | **821s** | **2.17×** |

itative visual comparisons of videos generated by SVOO in Figure 6, which further corroborate the quantitative results and demonstrate that our proposed SVOO still maintains great generation quality under high inference acceleration.

## 5.3. Ablation Study

To assess the contributions of the two mechanisms of SVOO to the quality-efficiency trade-off, we perform ablation studies on text-to-video generation using Wan2.1-1.3B, Wan2.2-14B, and HunyuanVideo (HY). We consider two variants of SVOO: (1) SVOO (w/o Off), which removes offline profiling and uses a fixed recall threshold of $\tau = 90\%$; (2) SVOO (w/o On), which removes bidirectional co-clustering and adopts independent clustering. As shown in Figure 5, removing the offline stage reduces efficiency, while removing the online stage degrades quality, demonstrating that offline profiling enables safe acceleration and online co-clustering yields better-aligned blocks with just minor extra overhead.

## 5.4. Quality-Efficiency Trade-off Study

We study the quality-efficiency trade-off of our proposed SVOO under different sparsity settings on a random subset of VBench. As shown in Figure 7, although the generation quality gradually decreases as sparsity increases and

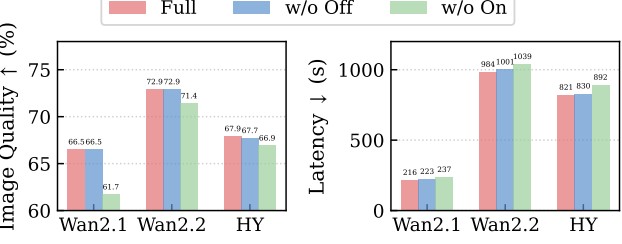

*Figure 5.* The ablation study of our proposed SVOO.

higher speedup is achieved, SVOO consistently maintains strong performance across a wide range of sparsity levels, demonstrating its robustness under aggressive acceleration.

## 5.5. Clustering Result Reuse Study

We analyze our co-clustering results to justify its reuse during inference. As shown in Figure 8, the mutual-information similarity of clustering results remains high across diffusion steps, indicating stable block partitions. Such stability enables reuse with recomputation every few steps and negligible quality loss, while substantially reducing the overhead.

## 5.6. Clustering Strategy Results Compare to SVG2

To analyze the importance of Q-K coupled block partitioning, we compare bidirectional co-clustering with the K-

*Table 2.* Quality and efficiency benchmarking results of our proposed SVOO and baselines on Image-to-Video Task.

| Model | Config | Baseline | Quality | | | | | | | Efficiency | |
|---|---|---|---|---|---|---|---|---|---|---|---|
| | | | PSNR↑ | SSIM↑ | LPIPS↓ | ImageQual↑ | AesQual↑ | SubConsist↑ | BackConsist↑ | Latency | Speedup |
| Wan2.1 | 14B-720P-I2V | Origin | - | - | - | 71.52% | 61.86% | 94.98% | 95.37% | 1658s | 1.00× |
| | | SpargeAttn | 21.557 | 0.691 | 0.297 | 70.83% | **61.77%** | 95.81% | 95.04% | 1124s | 1.48× |
| | | SVG1 | 24.262 | 0.825 | 0.174 | 70.93% | 61.02% | 94.42% | 94.51% | 1047s | 1.58× |
| | | SVG2 | 27.324 | 0.856 | 0.125 | 70.76% | 61.45% | **95.72%** | 94.93% | 998s | 1.66× |
| | | Radial | 23.673 | 0.759 | 0.189 | 70.91% | 61.62% | 94.52% | 95.36% | 1046s | 1.58× |
| | | **SVOO** | **27.545** | **0.878** | **0.121** | **71.71%** | 61.67% | 94.80% | **95.39%** | **954s** | **1.74×** |
| Wan2.2 | 14B-720P-I2V | Origin | - | - | - | 74.36% | 66.23% | 97.55% | 97.24% | 1605s | 1.00× |
| | | SpargeAttn | 25.935 | 0.832 | 0.139 | 72.61% | 62.29% | 97.03% | **97.09%** | 1119s | 1.42× |
| | | SVG1 | 26.882 | 0.866 | 0.131 | 72.88% | 62.05% | 97.17% | 97.08% | 1034s | 1.55× |
| | | SVG2 | 28.384 | 0.893 | 0.106 | 71.28% | 62.36% | 96.79% | 96.65% | 1057s | 1.52× |
| | | Radial | 25.080 | 0.797 | 0.156 | 71.19% | 63.73% | 95.91% | 96.49% | 1157s | 1.39× |
| | | **SVOO** | **29.678** | **0.913** | **0.095** | **73.37%** | **63.76%** | **97.31%** | 96.97% | **994s** | **1.61×** |
| Hunyuan | 13B-720P-I2V | Origin | - | - | - | 70.30% | 62.06% | 96.55% | 96.54% | 1761s | 1.00× |
| | | SpargeAttn | 22.908 | 0.717 | 0.259 | 68.53% | 60.99% | 96.22% | 95.37% | 1287s | 1.37× |
| | | SVG1 | 23.437 | 0.729 | 0.236 | 67.04% | 55.57% | 96.51% | 95.47% | 887s | 1.98× |
| | | SVG2 | 24.947 | 0.761 | 0.220 | 68.45% | 59.75% | 95.61% | 95.68% | 889s | 1.98× |
| | | Radial | 23.463 | 0.720 | 0.258 | 69.52% | **61.66%** | 95.30% | 95.62% | 912s | 1.93× |
| | | **SVOO** | **25.155** | **0.759** | **0.200** | **69.70%** | 59.68% | **96.79%** | **95.70%** | **810s** | **2.17×** |

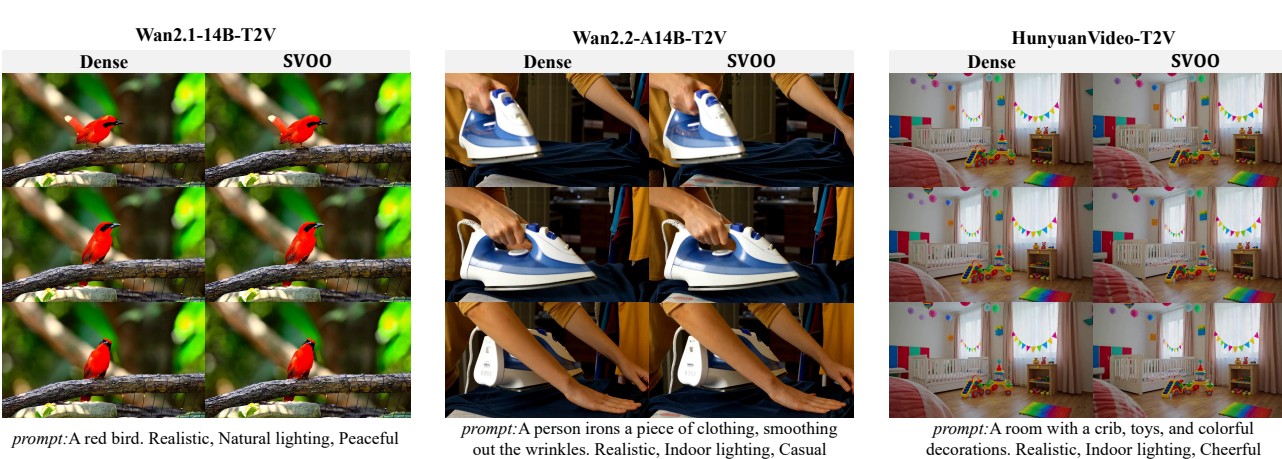

**Wan2.1-14B-T2V** · Dense · SVOO
*prompt:*A red bird. Realistic, Natural lighting, Peaceful

**Wan2.2-A14B-T2V** · Dense · SVOO
*prompt:*A person irons a piece of clothing, smoothing out the wrinkles. Realistic, Indoor lighting, Casual

**HunyuanVideo-T2V** · Dense · SVOO
*prompt:*A room with a crib, toys, and colorful decorations. Realistic, Indoor lighting, Cheerful

*Figure 6.* Examples of videos generated by our proposed SVOO and dense attention on Wan and HunyuanVideo models.

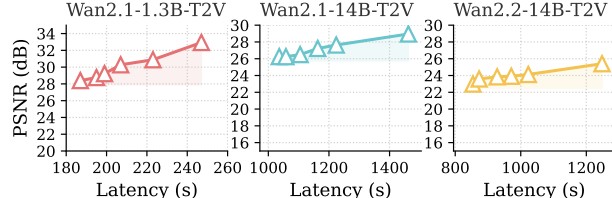

*Figure 7.* Quality-efficiency trade-off of SVOO under different sparsity settings. SVOO achieves higher speedup with only mild quality degradation as latency decreases, demonstrating a favorable trade-off between generation quality and inference efficiency.

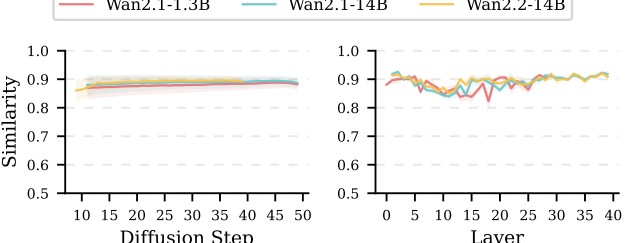

*Figure 8.* Analysis of clustering result consistency across diffusion steps and layers. The clustering assignments remain highly similar throughout the diffusion process and across most layers, indicating that the learned block partitions are stable and can be safely reused to reduce clustering overhead.

means method used in SVG2. We evaluate effectiveness using attention recall (details in Appendix B). As shown in Figure 9, our proposed bidirectional co-clustering consis-

tently achieves higher recall, demonstrating the benefit of

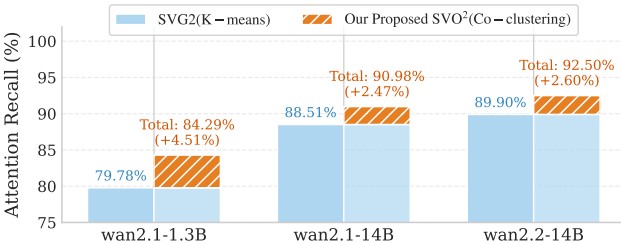

*Figure 9.* Attention recall comparison between SVOO and SVG2.

Q-K coupling over the independent Q-K block partitioning.

Further analyses of reuse steps, block numbers, and offline-stage computational overhead are provided in Appendix B.

## 6. Conclusion

We presented SVOO, a novel training-free sparse attention framework tailored for fast diffusion-based video generation that explicitly addresses two key limitations of prior methods: ignoring layer-wise heterogeneity in attention pruning and ignoring Q-K coupling in block partitioning. Our proposed SVOO address these limitations via a two-stage paradigm: (1) profiling layer-wise pruning tolerance offline to derive a reliable sparsity schedule, and (2) performing online bidirectional co-clustering to construct Q-K coupled, well-aligned blocks for more effective block-wise sparse attention. Extensive experiments on seven widely used video generation models demonstrate that our proposed SVOO consistently achieves a superior quality-efficiency trade-off.

## Impact Statement

The goal of this paper is to accelerate Diffusion Transformer-based video generation without requiring retraining. While this work may have various potential societal implications, we do not identify any that warrant specific discussion here.

## Acknowledgements

The corresponding authors are Jianxin Li and Zhibo Chen. This work was supported by the National Natural Science Foundation of China under Grant No. 62225202 and Grant No. 62302023, and the Zhongguancun Academy Project under Grant No. C20250302.

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

## A. Proof.

### A.1. Proof of Theorem 4.2

Here we first restate Theorem 4.2:

**Theorem A.1** (Layer-wise Sparsity Stability). *Consider a well-trained transformer layer and denote by $V(\mathbf{X})$ the average row-wise variance of the pre-softmax attention logits produced by this layer for input $\mathbf{X} \in \mathbb{R}^{n \times d}$. Under Assumption 4.1, for any two independent inputs $\mathbf{X}, \hat{\mathbf{X}} \in \mathbb{R}^{n \times d}$ of equal token length, it holds with probability at least $1 - \delta$:*

$$
\left| V(\mathbf{X}) - V(\hat{\mathbf{X}}) \right|
$$
$$
\leq \frac{d \, \|\mathbf{M}\|_2^2}{d'} C R^4 \left( \sqrt{\frac{\log(d/\delta)}{n}} + \frac{\log(d/\delta)}{n} \right), \tag{8}
$$

*where $C > 0$ is an absolute constant, $\mathbf{M} \triangleq \mathbf{W}_Q \mathbf{W}_K^\top$ with $\mathbf{W}_Q, \mathbf{W}_K \in \mathbb{R}^{d \times d'}$ are the query and key projection matrices of this layer, and $d'$ is the attention head dimension.*

*Proof.* **Attention Calculation.** Consider a transformer layer with input tensor $\mathbf{X} \in \mathbb{R}^{n \times d}$, where $n$ denotes the number of tokens and $d$ the hidden dimension. Let $\mathbf{W}_Q, \mathbf{W}_K \in \mathbb{R}^{d \times d'}$ be the projection matrices for queries and keys, respectively. The projected features are:

$$
\mathbf{Q} = \mathbf{X}\mathbf{W}_Q \in \mathbb{R}^{n \times d'}, \mathbf{K} = \mathbf{X}\mathbf{W}_K \in \mathbb{R}^{n \times d'}. \tag{9}
$$

The unnormalized scaled dot-product attention score matrix is then given by:

$$
\mathbf{A} = \frac{1}{\sqrt{d'}} \mathbf{Q}\mathbf{K}^\top = \frac{1}{\sqrt{d'}} \mathbf{X}\mathbf{W}_Q \mathbf{W}_K^\top \mathbf{X}^\top, \tag{10}
$$

where the second equality follows by substituting the definitions of $\mathbf{Q}$ and $\mathbf{K}$ and rearranging the matrix products.

**Variance Calculation.** For the $i$-th row of the attention matrix $\mathbf{A}$, we have:

$$
\mathbf{A}_{i,:} = \frac{1}{\sqrt{d'}} \mathbf{X}_{i,:} \mathbf{W}_Q \mathbf{W}_K^\top \mathbf{X}^\top, \tag{11}
$$

Let $\mu_i$ denote the mean of the $i$-th attention row, defined as:

$$
\mu_i = \frac{1}{n} \sum_{j=1}^{n} \mathbf{A}_{i,j} \tag{12}
$$

The variance of $\mathbf{A}_{i,:}$ the given by:

$$
\begin{aligned}
\mathrm{Var}(\mathbf{A}_{i,:}) &= \frac{1}{n} \sum_{j=1}^{n} (\mathbf{A}_{i,:} - \mu_i)^2 \\
&= \frac{1}{n} \sum_{j=1}^{n} (\mathbf{A}_{i,j}^2 - 2\mu_i \mathbf{A}_{i,j} + \mu_i^2) \\
&= \frac{1}{n} \sum_{j=1}^{n} \mathbf{A}_{i,j}^2 - 2\mu_i \left( \frac{1}{n} \sum_{j=1}^{n} \mathbf{A}_{i,j} \right) + \frac{1}{n} \sum_{j=1}^{n} \mu_i^2 \\
&= \frac{1}{n} \sum_{j=1}^{n} \mathbf{A}_{i,j}^2 - 2\mu_i^2 + \mu_i^2 \\
&= \frac{1}{n} \sum_{j=1}^{n} \mathbf{A}_{i,j}^2 - \mu_i^2
\end{aligned} \tag{13}
$$

We first compute the first term $\sum_{j=1}^{n} \mathbf{A}_{i,j}^2$. By definition,

$$
\sum_{j=1}^{n} \mathbf{A}_{i,j}^2 = \|\mathbf{A}_{i,:}\|_2^2 = \mathbf{A}_{i,:} \mathbf{A}_{i,:}^\top. \tag{14}
$$

Substituting $\mathbf{A}_{i,:} = \frac{1}{\sqrt{d'}}\mathbf{X}_{i,:}\mathbf{W}_\mathrm{Q}\mathbf{W}_\mathrm{K}^\top\mathbf{X}^\top$ yields

$$\mathbf{A}_{i,:}\mathbf{A}_{i,:}^\top = \frac{1}{d'}\,\mathbf{X}_{i,:}\mathbf{W}_\mathrm{Q}\mathbf{W}_\mathrm{K}^\top\mathbf{X}^\top\mathbf{X}\mathbf{W}_\mathrm{K}\mathbf{W}_\mathrm{Q}^\top\mathbf{X}_{i,:}^\top. \tag{15}$$

For notational simplicity, define

$$\mathbf{M} \triangleq \mathbf{W}_\mathrm{Q}\mathbf{W}_\mathrm{K}^\top. \tag{16}$$

Then we can write

$$\sum_{j=1}^{n}\mathbf{A}_{i,j}^2 = \frac{1}{d'}\,\mathbf{X}_{i,:}\mathbf{M}\,\mathbf{X}^\top\mathbf{X}\,\mathbf{M}^\top\mathbf{X}_{i,:}^\top. \tag{17}$$

Next, we compute the second term $\mu_i^2$. By definition, the mean of the $i$-th attention row is given by

$$
\begin{aligned}
\mu_i &= \frac{1}{n}\sum_{j=1}^{n}\mathbf{A}_{i,j}\\
&= \frac{1}{n}\sum_{j=1}^{n}\frac{1}{\sqrt{d'}}\mathbf{X}_{i,:}\mathbf{W}_\mathrm{Q}\mathbf{W}_\mathrm{K}^\top\mathbf{X}_{j,:}^\top\\
&= \frac{1}{\sqrt{d'}}\,\mathbf{X}_{i,:}\mathbf{W}_\mathrm{Q}\mathbf{W}_\mathrm{K}^\top\left(\frac{1}{n}\sum_{j=1}^{n}\mathbf{X}_{j,:}^\top\right)\\
&= \frac{1}{\sqrt{d'}}\,\mathbf{X}_{i,:}\mathbf{W}_\mathrm{Q}\mathbf{W}_\mathrm{K}^\top\boldsymbol{\mu}^\top,\\
&= \frac{1}{\sqrt{d'}}\,\mathbf{X}_{i,:}\mathbf{M}\boldsymbol{\mu}^\top
\end{aligned}
\tag{18}
$$

where $\boldsymbol{\mu} \in \mathbb{R}^{1\times d}$ denotes the mean token representation,

$$\boldsymbol{\mu} = \frac{1}{n}\mathbf{1}^\top\mathbf{X}. \tag{19}$$

Thus, combining Eq. (13), Eq. (18), and Eq. (17), we obtain:

$$
\begin{aligned}
\mathrm{Var}(\mathbf{A}_{i,:}) &= \frac{1}{n}\sum_{j=1}^{n}\mathbf{A}_{i,j}^2 - \mu_i^2\\
&= \frac{1}{n}(\frac{1}{d'}\,\mathbf{X}_{i,:}\mathbf{M}\,\mathbf{X}^\top\mathbf{X}\,\mathbf{M}^\top\mathbf{X}_{i,:}^\top) - (\frac{1}{\sqrt{d'}}\,\mathbf{X}_{i,:}\mathbf{M}\boldsymbol{\mu}^\top)^2\\
&= \frac{1}{n}\frac{1}{d'}\,\mathbf{X}_{i,:}\mathbf{M}\,\mathbf{X}^\top\mathbf{X}\,\mathbf{M}^\top\mathbf{X}_{i,:}^\top - \frac{1}{d'}\,\mathbf{X}_{i,:}\mathbf{M}\boldsymbol{\mu}^\top\boldsymbol{\mu}\mathbf{M}^\top\mathbf{X}_{i,:}^\top\\
&= \frac{1}{d'}\mathbf{X}_{i,:}\mathbf{M}(\frac{1}{n}\mathbf{X}^\top\mathbf{X} - \boldsymbol{\mu}^\top\boldsymbol{\mu})\mathbf{M}^\top\mathbf{X}_{i,:}^\top.
\end{aligned}
\tag{20}
$$

We now relate the above expression to the covariance matrix of the input representations. The sample covariance matrix $\boldsymbol{\Sigma}$ of $\mathbf{X}$ is defined as:

$$
\begin{aligned}
\boldsymbol{\Sigma} &= \frac{1}{n}(\mathbf{X} - \mathbf{1}\boldsymbol{\mu})^\top(\mathbf{X} - \mathbf{1}\boldsymbol{\mu})\\
&= \frac{1}{n}\mathbf{X}^\top\mathbf{X} - \boldsymbol{\mu}^\top\mathbf{1}^\top\mathbf{X} - \mathbf{X}^\top\mathbf{1}\boldsymbol{\mu} + \boldsymbol{\mu}^\top\mathbf{1}^\top\mathbf{1}\boldsymbol{\mu}\\
&= \frac{1}{n}\mathbf{X}^\top\mathbf{X} - \boldsymbol{\mu}^\top n\boldsymbol{\mu} - n\boldsymbol{\mu}^\top\boldsymbol{\mu} + \boldsymbol{\mu}^\top\mathbf{1}^\top\mathbf{1}\boldsymbol{\mu}\\
&= \frac{1}{n}\mathbf{X}^\top\mathbf{X} - n\boldsymbol{\mu}^\top\boldsymbol{\mu} - n\boldsymbol{\mu}^\top\boldsymbol{\mu} + n\boldsymbol{\mu}^\top\boldsymbol{\mu}\\
&= \frac{1}{n}(\mathbf{X}^\top\mathbf{X} - n\boldsymbol{\mu}^\top\boldsymbol{\mu})\\
&= \frac{1}{n}\mathbf{X}^\top\mathbf{X} - \boldsymbol{\mu}^\top\boldsymbol{\mu},
\end{aligned}
\tag{21}
$$

where $\mathbf{1} \in \mathbb{R}^n$ denotes the all-ones vector and $\boldsymbol{\mu} = \frac{1}{n}\mathbf{1}^\top \mathbf{X}$ is the mean token representation. Substituting this definition into Eq. (20), we can rewrite the attention variance as:

$$\begin{aligned}
\mathrm{Var}(\mathbf{A}_{i,:}) &= \frac{1}{d'}\mathbf{X}_{i,:}\mathbf{M}(\frac{1}{n}\mathbf{X}^\top \mathbf{X} - \boldsymbol{\mu}^\top \boldsymbol{\mu})\mathbf{M}^\top \mathbf{X}_{i,:}^\top \\
&= \frac{1}{d'}\mathbf{X}_{i,:}\mathbf{M}\boldsymbol{\Sigma}\mathbf{M}^\top \mathbf{X}_{i,:}^\top
\end{aligned} \tag{22}$$

Since the softmax normalization is applied independently to each row of the attention score matrix $\mathbf{A}$, the sparsity of attention can be characterized by the variability within each attention row. We therefore consider the average row-wise variance,

$$\mathbb{E}[\mathrm{Var}(\mathbf{A}_{i,:})] \triangleq \frac{1}{n}\sum_{i=1}^{n}\mathrm{Var}(\mathbf{A}_{i,:}), \tag{23}$$

which serves as a proxy for attention sparsity. Substituting the expression derived above yields

$$\mathbb{E}[\mathrm{Var}(\mathbf{A}_{i,:})] = \frac{1}{n}\sum_{i=1}^{n}\left(\frac{1}{d'}\mathbf{X}_{i,:}\mathbf{M}\boldsymbol{\Sigma}\mathbf{M}^\top \mathbf{X}_{i,:}^\top\right). \tag{24}$$

Next, we rewrite the summation in a compact trace form. Let $\mathbf{B} \triangleq \mathbf{M}\boldsymbol{\Sigma}\mathbf{M}^\top \in \mathbb{R}^{d\times d}$. Then,

$$\begin{aligned}
\frac{1}{n}\sum_{i=1}^{n}\mathbf{X}_{i,:}\mathbf{B}\mathbf{X}_{i,:}^\top &= \frac{1}{n}\sum_{i=1}^{n}\mathrm{tr}\left(\mathbf{X}_{i,:}\mathbf{B}\mathbf{X}_{i,:}^\top\right) \\
&= \frac{1}{n}\sum_{i=1}^{n}\mathrm{tr}\left(\mathbf{B}\mathbf{X}_{i,:}^\top \mathbf{X}_{i,:}\right) \\
&= \mathrm{tr}\left(\mathbf{B}\cdot\frac{1}{n}\sum_{i=1}^{n}\mathbf{X}_{i,:}^\top \mathbf{X}_{i,:}\right) \\
&= \mathrm{tr}\left(\mathbf{B}\cdot\frac{1}{n}\mathbf{X}^\top \mathbf{X}\right),
\end{aligned} \tag{25}$$

where we used $\sum_{i=1}^{n}\mathbf{X}_{i,:}^\top \mathbf{X}_{i,:} = \mathbf{X}^\top \mathbf{X}$.

Therefore,

$$\mathbb{E}[\mathrm{Var}(\mathbf{A}_{i,:})] = \frac{1}{d'}\mathrm{tr}\left(\mathbf{M}\boldsymbol{\Sigma}\mathbf{M}^\top \cdot \frac{1}{n}\mathbf{X}^\top \mathbf{X}\right). \tag{26}$$

Finally, using the identity

$$\boldsymbol{\Sigma} = \frac{1}{n}\mathbf{X}^\top \mathbf{X} - \boldsymbol{\mu}^\top \boldsymbol{\mu} \quad \Longleftrightarrow \quad \frac{1}{n}\mathbf{X}^\top \mathbf{X} = \boldsymbol{\Sigma} + \boldsymbol{\mu}^\top \boldsymbol{\mu}, \tag{27}$$

we obtain

$$\boxed{\mathbb{E}[\mathrm{Var}(\mathbf{A}_{i,:})] = \frac{1}{d'}\mathrm{tr}\left(\mathbf{M}\boldsymbol{\Sigma}\mathbf{M}^\top \left(\boldsymbol{\Sigma} + \boldsymbol{\mu}^\top \boldsymbol{\mu}\right)\right).} \tag{28}$$

**Setup.** Let $\mathbf{X}, \widehat{\mathbf{X}} \in \mathbb{R}^{n\times d}$ be two independent samples. Denote rows by $\mathbf{x}_i = \mathbf{X}_{i,:} \in \mathbb{R}^{1\times d}$ and $\widehat{\mathbf{x}}_i = \widehat{\mathbf{X}}_{i,:} \in \mathbb{R}^{1\times d}$. Define the sample means

$$\boldsymbol{\mu} \triangleq \frac{1}{n}\sum_{i=1}^{n}\mathbf{x}_i \in \mathbb{R}^{1\times d}, \qquad \widehat{\boldsymbol{\mu}} \triangleq \frac{1}{n}\sum_{i=1}^{n}\widehat{\mathbf{x}}_i \in \mathbb{R}^{1\times d}, \tag{29}$$

and the (row-wise) sample covariance matrices

$$\boldsymbol{\Sigma} \triangleq \frac{1}{n}\sum_{i=1}^{n}(\mathbf{x}_i - \boldsymbol{\mu})^\top(\mathbf{x}_i - \boldsymbol{\mu}) \in \mathbb{R}^{d\times d}, \qquad \widehat{\boldsymbol{\Sigma}} \triangleq \frac{1}{n}\sum_{i=1}^{n}(\widehat{\mathbf{x}}_i - \widehat{\boldsymbol{\mu}})^\top(\widehat{\mathbf{x}}_i - \widehat{\boldsymbol{\mu}}) \in \mathbb{R}^{d\times d}. \tag{30}$$

Recall $\mathbf{M} \triangleq \mathbf{W}_Q \mathbf{W}_K^\top$ and define

$$\mathbf{G} \triangleq \mathbf{M}^\top \mathbf{M} \succeq 0, \qquad \|\mathbf{G}\|_2 = \|\mathbf{M}\|_2^2. \tag{31}$$

As shown previously, the average row-wise variance of attention logits (pre-softmax) equals

$$V(\mathbf{X}) \triangleq \frac{1}{n} \sum_{i=1}^n \mathrm{Var}(\mathbf{A}_{i,:}) = \frac{1}{d'} \mathrm{tr}\Big(\mathbf{M}\boldsymbol{\Sigma}\mathbf{M}^\top\big(\boldsymbol{\Sigma} + \boldsymbol{\mu}^\top \boldsymbol{\mu}\big)\Big). \tag{32}$$

Using the cyclic property of trace,

$$V(\mathbf{X}) = \frac{1}{d'} \mathrm{tr}\Big(\mathbf{G}\,\boldsymbol{\Sigma}\,(\boldsymbol{\Sigma} + \boldsymbol{\mu}^\top \boldsymbol{\mu})\Big), \qquad V(\widehat{\mathbf{X}}) = \frac{1}{d'} \mathrm{tr}\Big(\mathbf{G}\,\widehat{\boldsymbol{\Sigma}}\,(\widehat{\boldsymbol{\Sigma}} + \widehat{\boldsymbol{\mu}}^\top \widehat{\boldsymbol{\mu}})\Big). \tag{33}$$

**Step 1: Deterministic bound on** $\big|V(\mathbf{X}) - V(\widehat{\mathbf{X}})\big|$**.** Let $\Delta_\Sigma \triangleq \boldsymbol{\Sigma} - \widehat{\boldsymbol{\Sigma}}$ and $\Delta_\mu \triangleq \boldsymbol{\mu} - \widehat{\boldsymbol{\mu}}$. Start from the difference:

$$\begin{aligned}
V(\mathbf{X}) - V(\widehat{\mathbf{X}}) &= \frac{1}{d'} \mathrm{tr}\Big(\mathbf{G}\,\boldsymbol{\Sigma}\,(\boldsymbol{\Sigma} + \boldsymbol{\mu}^\top \boldsymbol{\mu}) - \mathbf{G}\,\widehat{\boldsymbol{\Sigma}}\,(\widehat{\boldsymbol{\Sigma}} + \widehat{\boldsymbol{\mu}}^\top \widehat{\boldsymbol{\mu}})\Big) \\
&= \frac{1}{d'} \mathrm{tr}\Big(\mathbf{G}\big[\boldsymbol{\Sigma}(\boldsymbol{\Sigma} + \boldsymbol{\mu}^\top \boldsymbol{\mu}) - \widehat{\boldsymbol{\Sigma}}(\widehat{\boldsymbol{\Sigma}} + \widehat{\boldsymbol{\mu}}^\top \widehat{\boldsymbol{\mu}})\big]\Big).
\end{aligned} \tag{34}$$

Now decompose the bracket term by adding and subtracting $\widehat{\boldsymbol{\Sigma}}(\boldsymbol{\Sigma} + \boldsymbol{\mu}^\top \boldsymbol{\mu})$:

$$\begin{aligned}
&\boldsymbol{\Sigma}(\boldsymbol{\Sigma} + \boldsymbol{\mu}^\top \boldsymbol{\mu}) - \widehat{\boldsymbol{\Sigma}}(\widehat{\boldsymbol{\Sigma}} + \widehat{\boldsymbol{\mu}}^\top \widehat{\boldsymbol{\mu}}) \\
&= \Big(\boldsymbol{\Sigma}(\boldsymbol{\Sigma} + \boldsymbol{\mu}^\top \boldsymbol{\mu}) - \widehat{\boldsymbol{\Sigma}}(\boldsymbol{\Sigma} + \boldsymbol{\mu}^\top \boldsymbol{\mu})\Big) + \Big(\widehat{\boldsymbol{\Sigma}}(\boldsymbol{\Sigma} + \boldsymbol{\mu}^\top \boldsymbol{\mu}) - \widehat{\boldsymbol{\Sigma}}(\widehat{\boldsymbol{\Sigma}} + \widehat{\boldsymbol{\mu}}^\top \widehat{\boldsymbol{\mu}})\Big) \\
&= \Delta_\Sigma(\boldsymbol{\Sigma} + \boldsymbol{\mu}^\top \boldsymbol{\mu}) + \widehat{\boldsymbol{\Sigma}}\Big((\boldsymbol{\Sigma} - \widehat{\boldsymbol{\Sigma}}) + (\boldsymbol{\mu}^\top \boldsymbol{\mu} - \widehat{\boldsymbol{\mu}}^\top \widehat{\boldsymbol{\mu}})\Big) \\
&= \Delta_\Sigma(\boldsymbol{\Sigma} + \boldsymbol{\mu}^\top \boldsymbol{\mu}) + \widehat{\boldsymbol{\Sigma}}\Delta_\Sigma + \widehat{\boldsymbol{\Sigma}}(\boldsymbol{\mu}^\top \boldsymbol{\mu} - \widehat{\boldsymbol{\mu}}^\top \widehat{\boldsymbol{\mu}}).
\end{aligned} \tag{35}$$

Plugging (35) into (34) and using triangle inequality gives

$$\begin{aligned}
\big|V(\mathbf{X}) - V(\widehat{\mathbf{X}})\big| \leq \frac{1}{d'} \Big(&\big|\mathrm{tr}\big(\mathbf{G}\,\Delta_\Sigma(\boldsymbol{\Sigma} + \boldsymbol{\mu}^\top \boldsymbol{\mu})\big)\big| + \big|\mathrm{tr}\big(\mathbf{G}\,\widehat{\boldsymbol{\Sigma}}\,\Delta_\Sigma\big)\big| \\
&+ \big|\mathrm{tr}\big(\mathbf{G}\,\widehat{\boldsymbol{\Sigma}}(\boldsymbol{\mu}^\top \boldsymbol{\mu} - \widehat{\boldsymbol{\mu}}^\top \widehat{\boldsymbol{\mu}})\big)\big|\Big).
\end{aligned} \tag{36}$$

Next we upper bound each trace term. We use the inequality

$$|\mathrm{tr}(\mathbf{U}\mathbf{V})| \leq \|\mathbf{U}\|_2 \|\mathbf{V}\|_*, \tag{37}$$

and the sub-multiplicativity of nuclear norm:

$$\|\mathbf{A}\mathbf{B}\|_* \leq \|\mathbf{A}\|_2 \|\mathbf{B}\|_*, \qquad \|\mathbf{B}\|_* \leq d\,\|\mathbf{B}\|_2. \tag{38}$$

Applying (37) to each term in (36) gives

$$\begin{aligned}
\big|V(\mathbf{X}) - V(\widehat{\mathbf{X}})\big| &\leq \frac{1}{d'} \|\mathbf{G}\|_2 \Big(\|\Delta_\Sigma(\boldsymbol{\Sigma} + \boldsymbol{\mu}^\top \boldsymbol{\mu})\|_* + \|\widehat{\boldsymbol{\Sigma}}\,\Delta_\Sigma\|_* + \|\widehat{\boldsymbol{\Sigma}}(\boldsymbol{\mu}^\top \boldsymbol{\mu} - \widehat{\boldsymbol{\mu}}^\top \widehat{\boldsymbol{\mu}})\|_*\Big) \\
&= \frac{\|\mathbf{M}\|_2^2}{d'} \Big(\|\Delta_\Sigma(\boldsymbol{\Sigma} + \boldsymbol{\mu}^\top \boldsymbol{\mu})\|_* + \|\widehat{\boldsymbol{\Sigma}}\,\Delta_\Sigma\|_* + \|\widehat{\boldsymbol{\Sigma}}(\boldsymbol{\mu}^\top \boldsymbol{\mu} - \widehat{\boldsymbol{\mu}}^\top \widehat{\boldsymbol{\mu}})\|_*\Big).
\end{aligned} \tag{39}$$

Using (38), we further obtain an operator-norm-only bound:

$$\begin{aligned}
\big|V(\mathbf{X}) - V(\widehat{\mathbf{X}})\big| &\leq \frac{\|\mathbf{M}\|_2^2}{d'} \Big(\|\Delta_\Sigma\|_2 \|\boldsymbol{\Sigma} + \boldsymbol{\mu}^\top \boldsymbol{\mu}\|_* + \|\widehat{\boldsymbol{\Sigma}}\|_2 \|\Delta_\Sigma\|_* + \|\widehat{\boldsymbol{\Sigma}}\|_2 \|\boldsymbol{\mu}^\top \boldsymbol{\mu} - \widehat{\boldsymbol{\mu}}^\top \widehat{\boldsymbol{\mu}}\|_*\Big) \\
&\leq \frac{d\,\|\mathbf{M}\|_2^2}{d'} \Big(\|\Delta_\Sigma\|_2 \|\boldsymbol{\Sigma} + \boldsymbol{\mu}^\top \boldsymbol{\mu}\|_2 + \|\widehat{\boldsymbol{\Sigma}}\|_2 \|\Delta_\Sigma\|_2 + \|\widehat{\boldsymbol{\Sigma}}\|_2 \|\boldsymbol{\mu}^\top \boldsymbol{\mu} - \widehat{\boldsymbol{\mu}}^\top \widehat{\boldsymbol{\mu}}\|_2\Big).
\end{aligned} \tag{40}$$

Finally we bound the rank-one difference term. Note that

$$\boldsymbol{\mu}^\top \boldsymbol{\mu} - \widehat{\boldsymbol{\mu}}^\top \widehat{\boldsymbol{\mu}} = (\boldsymbol{\mu} - \widehat{\boldsymbol{\mu}})^\top \boldsymbol{\mu} + \widehat{\boldsymbol{\mu}}^\top (\boldsymbol{\mu} - \widehat{\boldsymbol{\mu}}),$$ (41)

so by the triangle inequality and $\|\mathbf{a}^\top \mathbf{b}\|_2 = \|\mathbf{a}\|_2 \|\mathbf{b}\|_2$ for rank-one outer products,

$$
\begin{aligned}
\|\boldsymbol{\mu}^\top \boldsymbol{\mu} - \widehat{\boldsymbol{\mu}}^\top \widehat{\boldsymbol{\mu}}\|_2 &\leq \|(\boldsymbol{\mu} - \widehat{\boldsymbol{\mu}})^\top \boldsymbol{\mu}\|_2 + \|\widehat{\boldsymbol{\mu}}^\top (\boldsymbol{\mu} - \widehat{\boldsymbol{\mu}})\|_2 \\
&= \|\Delta_\mu\|_2 \|\boldsymbol{\mu}\|_2 + \|\widehat{\boldsymbol{\mu}}\|_2 \|\Delta_\mu\|_2 \\
&= (\|\boldsymbol{\mu}\|_2 + \|\widehat{\boldsymbol{\mu}}\|_2) \|\Delta_\mu\|_2.
\end{aligned}
$$ (42)

Combining (40) and (42) yields a fully deterministic bound:

$$\left| V(\mathbf{X}) - V(\widehat{\mathbf{X}}) \right| \leq \frac{d \|\mathbf{M}\|_2^2}{d'} \left( \|\Delta_\Sigma\|_2 \|\boldsymbol{\Sigma} + \boldsymbol{\mu}^\top \boldsymbol{\mu}\|_2 + \|\widehat{\boldsymbol{\Sigma}}\|_2 \|\Delta_\Sigma\|_2 + \|\widehat{\boldsymbol{\Sigma}}\|_2 (\|\boldsymbol{\mu}\|_2 + \|\widehat{\boldsymbol{\mu}}\|_2) \|\Delta_\mu\|_2 \right).$$ (43)

Here we restate the Assumption.

**Assumption A.2** (Bounded Token Representations). Consider a transformer layer and let its input be $\mathbf{X} \in \mathbb{R}^{n \times d}$, whose rows are token representations $\mathbf{x}_i \in \mathbb{R}^{1 \times d}$. Assume $\{\mathbf{x}_i\}_{i=1}^n$ are from a distribution on $\mathbb{R}^d$ with population mean $\boldsymbol{\mu}_\star \triangleq \mathbb{E}[\mathbf{x}]$ and covariance $\boldsymbol{\Sigma}_\star \triangleq \mathbb{E}\left[(\mathbf{x} - \boldsymbol{\mu}_\star)^\top (\mathbf{x} - \boldsymbol{\mu}_\star)\right]$. We assume that there exists a constant $R > 0$ such that:

$$\|\mathbf{x}\|_2 \leq R.$$ (44)

**Step 2: High-probability bounds under bounded layer inputs.**

Assume Assumption 4.1 holds, i.e., $\|\mathbf{x}\|_2 \leq R$ almost surely. Let $\boldsymbol{\mu}_\star \triangleq \mathbb{E}[\mathbf{x}]$ and $\boldsymbol{\Sigma}_\star \triangleq \mathbb{E}\left[(\mathbf{x} - \boldsymbol{\mu}_\star)^\top (\mathbf{x} - \boldsymbol{\mu}_\star)\right]$.

**Step 2.1: Basic consequences of bounded inputs.** By Jensen's inequality,

$$\|\boldsymbol{\mu}_\star\|_2 = \|\mathbb{E}[\mathbf{x}]\|_2 \leq \mathbb{E}\|\mathbf{x}\|_2 \leq R.$$ (45)

Moreover, for any unit vector $\mathbf{u} \in \mathbb{R}^d$,

$$\mathbf{u}^\top \boldsymbol{\Sigma}_\star \mathbf{u} = \mathbb{E}\left[(\mathbf{u}^\top (\mathbf{x} - \boldsymbol{\mu}_\star))^2\right] \leq \mathbb{E}\|\mathbf{x} - \boldsymbol{\mu}_\star\|_2^2 \leq (\|\mathbf{x}\|_2 + \|\boldsymbol{\mu}_\star\|_2)^2 \leq (2R)^2,$$ (46)

hence

$$\|\boldsymbol{\Sigma}_\star\|_2 \leq 4R^2.$$ (47)

**Step 2.2: Concentration of the sample mean.** Define the sample mean

$$\boldsymbol{\mu} \triangleq \frac{1}{n} \sum_{i=1}^n \mathbf{x}_i, \qquad \widehat{\boldsymbol{\mu}} \triangleq \frac{1}{n} \sum_{i=1}^n \widehat{\mathbf{x}}_i.$$

Let $\mathbf{z}_i \triangleq \mathbf{x}_i - \boldsymbol{\mu}_\star$ and $\widehat{\mathbf{z}}_i \triangleq \widehat{\mathbf{x}}_i - \boldsymbol{\mu}_\star$. Then

$$\boldsymbol{\mu} - \boldsymbol{\mu}_\star = \frac{1}{n} \sum_{i=1}^n \mathbf{z}_i, \qquad \widehat{\boldsymbol{\mu}} - \boldsymbol{\mu}_\star = \frac{1}{n} \sum_{i=1}^n \widehat{\mathbf{z}}_i.$$

Since $\|\mathbf{x}_i\|_2 \leq R$ and (45) holds,

$$\|\mathbf{z}_i\|_2 = \|\mathbf{x}_i - \boldsymbol{\mu}_\star\|_2 \leq \|\mathbf{x}_i\|_2 + \|\boldsymbol{\mu}_\star\|_2 \leq 2R \quad \text{a.s.},$$ (48)

and the same bound holds for $\widehat{\mathbf{z}}_i$.

By the vector Hoeffding inequality, for any $\delta \in (0, 1)$, with probability at least $1 - \delta$,

$$\|\boldsymbol{\mu} - \boldsymbol{\mu}_\star\|_2 \leq 2R\sqrt{\frac{2\log(2/\delta)}{n}}, \qquad \|\widehat{\boldsymbol{\mu}} - \boldsymbol{\mu}_\star\|_2 \leq 2R\sqrt{\frac{2\log(2/\delta)}{n}}.$$ (49)

Consequently, on the same event,

$$\|\Delta_\mu\|_2 = \|\boldsymbol{\mu} - \widehat{\boldsymbol{\mu}}\|_2 \leq 4R\sqrt{\frac{2\log(2/\delta)}{n}}.$$ (50)

**Step 2.3: Concentration of the sample covariance.** Define the sample covariance matrix

$$\mathbf{\Sigma} \triangleq \frac{1}{n} \sum_{i=1}^{n} (\mathbf{x}_i - \boldsymbol{\mu})^\top (\mathbf{x}_i - \boldsymbol{\mu}),$$

and define the oracle covariance

$$\widetilde{\mathbf{\Sigma}} \triangleq \frac{1}{n} \sum_{i=1}^{n} (\mathbf{x}_i - \boldsymbol{\mu}_\star)^\top (\mathbf{x}_i - \boldsymbol{\mu}_\star).$$

Using $\mathbf{x}_i - \boldsymbol{\mu} = (\mathbf{x}_i - \boldsymbol{\mu}_\star) - (\boldsymbol{\mu} - \boldsymbol{\mu}_\star)$, one checks that

$$\mathbf{\Sigma} = \widetilde{\mathbf{\Sigma}} - (\boldsymbol{\mu} - \boldsymbol{\mu}_\star)^\top (\boldsymbol{\mu} - \boldsymbol{\mu}_\star). \tag{51}$$

Let

$$\mathbf{Y}_i \triangleq (\mathbf{x}_i - \boldsymbol{\mu}_\star)^\top (\mathbf{x}_i - \boldsymbol{\mu}_\star) - \mathbf{\Sigma}_\star, \qquad \mathbb{E}[\mathbf{Y}_i] = \mathbf{0}.$$

Then

$$\widetilde{\mathbf{\Sigma}} - \mathbf{\Sigma}_\star = \frac{1}{n} \sum_{i=1}^{n} \mathbf{Y}_i.$$

Since $\|\mathbf{x}_i - \boldsymbol{\mu}_\star\|_2 \leq 2R$,

$$\|\mathbf{Y}_i\|_2 \leq \|\mathbf{x}_i - \boldsymbol{\mu}_\star\|_2^2 + \|\mathbf{\Sigma}_\star\|_2 \leq 4R^2 + 4R^2 = 8R^2.$$

Moreover,

$$\left\| \sum_{i=1}^{n} \mathbb{E}[\mathbf{Y}_i^2] \right\|_2 \leq n \cdot (8R^2)^2 = 64nR^4.$$

Applying the matrix Bernstein inequality yields that, with probability at least $1 - \delta$,

$$\|\widetilde{\mathbf{\Sigma}} - \mathbf{\Sigma}_\star\|_2 \leq C_1 R^2 \left( \sqrt{\frac{\log(2d/\delta)}{n}} + \frac{\log(2d/\delta)}{n} \right), \tag{52}$$

for an absolute constant $C_1 > 0$.

Combining (51), (49), and (52), we conclude that, with probability at least $1 - \delta$,

$$\|\mathbf{\Sigma} - \mathbf{\Sigma}_\star\|_2 \leq C_2 R^2 \left( \sqrt{\frac{\log(2d/\delta)}{n}} + \frac{\log(2d/\delta)}{n} \right), \tag{53}$$

for an absolute constant $C_2 > 0$. The same bound holds for $\widehat{\mathbf{\Sigma}}$.

Finally, by the triangle inequality,

$$\|\Delta_\Sigma\|_2 \leq C_3 R^2 \left( \sqrt{\frac{\log(2d/\delta)}{n}} + \frac{\log(2d/\delta)}{n} \right), \tag{54}$$

for an absolute constant $C_3 > 0$.

**Step 3: Plug-in and conclude the bound.**

Recall the deterministic inequality from Step 1:

$$\left| V(\mathbf{X}) - V(\widehat{\mathbf{X}}) \right| \leq \frac{d \|\mathbf{M}\|_2^2}{d'} \left( \|\Delta_\Sigma\|_2 \|\mathbf{\Sigma} + \boldsymbol{\mu}^\top \boldsymbol{\mu}\|_2 + \|\widehat{\mathbf{\Sigma}}\|_2 \|\Delta_\Sigma\|_2 + \|\widehat{\mathbf{\Sigma}}\|_2 (\|\boldsymbol{\mu}\|_2 + \|\widehat{\boldsymbol{\mu}}\|_2) \|\Delta_\mu\|_2 \right). \tag{55}$$

**Step 3.1: Define the high-probability event.** From Step 2, the bounds (50) and (54) each hold with probability at least $1 - \delta/2$ (by replacing $\delta$ with $\delta/2$ in Step 2). Let $\mathcal{E}$ denote the intersection of these two events:

$$\mathcal{E} \triangleq \left\{ \|\Delta_\mu\|_2 \leq 4R\sqrt{\frac{2\log(4/\delta)}{n}} \right\} \cap \left\{ \|\Delta_\Sigma\|_2 \leq C_3 R^2 \left( \sqrt{\frac{\log(4d/\delta)}{n}} + \frac{\log(4d/\delta)}{n} \right) \right\}. \tag{56}$$

By the union bound, $\mathbb{P}(\mathcal{E}) \geq 1 - \delta$. In the remainder of this step, we work on the event $\mathcal{E}$.

**Step 3.2: Deterministic bounds on $\|\mu\|_2$, $\|\widehat{\mu}\|_2$, $\|\Sigma\|_2$, and $\|\widehat{\Sigma}\|_2$.** Under Assumption 4.1, we have $\|\mathbf{x}_i\|_2 \leq R$ and $\|\widehat{\mathbf{x}}_i\|_2 \leq R$ almost surely.

**(a) Bound on sample means.** By triangle inequality and convexity,

$$\|\mu\|_2 = \left\| \frac{1}{n}\sum_{i=1}^{n} \mathbf{x}_i \right\|_2 \leq \frac{1}{n}\sum_{i=1}^{n} \|\mathbf{x}_i\|_2 \leq R, \qquad \|\widehat{\mu}\|_2 \leq \frac{1}{n}\sum_{i=1}^{n} \|\widehat{\mathbf{x}}_i\|_2 \leq R. \tag{57}$$

Consequently,

$$\|\mu\|_2 + \|\widehat{\mu}\|_2 \leq 2R. \tag{58}$$

**(b) Bound on sample covariances.** For each $i$, by (57) we have

$$\|\mathbf{x}_i - \mu\|_2 \leq \|\mathbf{x}_i\|_2 + \|\mu\|_2 \leq R + R = 2R. \tag{59}$$

Therefore,

$$\|\Sigma\|_2 = \left\| \frac{1}{n}\sum_{i=1}^{n} (\mathbf{x}_i - \mu)^\top (\mathbf{x}_i - \mu) \right\|_2 \leq \frac{1}{n}\sum_{i=1}^{n} \left\| (\mathbf{x}_i - \mu)^\top (\mathbf{x}_i - \mu) \right\|_2$$

$$= \frac{1}{n}\sum_{i=1}^{n} \|\mathbf{x}_i - \mu\|_2^2 \leq \frac{1}{n}\sum_{i=1}^{n} (2R)^2 = 4R^2. \tag{60}$$

The same argument gives

$$\|\widehat{\Sigma}\|_2 \leq 4R^2. \tag{61}$$

**Step 3.3: Bound $\|\Sigma + \mu^\top \mu\|_2$.** Using the triangle inequality and (57), (60),

$$\|\Sigma + \mu^\top \mu\|_2 \leq \|\Sigma\|_2 + \|\mu^\top \mu\|_2 = \|\Sigma\|_2 + \|\mu\|_2^2 \leq 4R^2 + R^2 = 5R^2. \tag{62}$$

**Step 3.4: Substitute bounds into (55).** Plugging (62), (61), and (58) into (55) yields

$$\left| V(\mathbf{X}) - V(\widehat{\mathbf{X}}) \right| \leq \frac{d\,\|\mathbf{M}\|_2^2}{d'} \left( \|\Delta_\Sigma\|_2 \cdot 5R^2 + (4R^2) \cdot \|\Delta_\Sigma\|_2 + (4R^2) \cdot (2R) \cdot \|\Delta_\mu\|_2 \right)$$

$$= \frac{d\,\|\mathbf{M}\|_2^2}{d'} \left( 9R^2\,\|\Delta_\Sigma\|_2 + 8R^3\,\|\Delta_\mu\|_2 \right). \tag{63}$$

**Step 3.5: Use the concentration bounds from Step 2.** On the event $\mathcal{E}$ in (56), substitute the bounds for $\|\Delta_\Sigma\|_2$ and $\|\Delta_\mu\|_2$ into (63). This gives, with probability at least $1 - \delta$,

$$\left| V(\mathbf{X}) - V(\widehat{\mathbf{X}}) \right| \leq \frac{d\,\|\mathbf{M}\|_2^2}{d'} \left[ 9R^2 \cdot C_3 R^2 \left( \sqrt{\frac{\log(4d/\delta)}{n}} + \frac{\log(4d/\delta)}{n} \right) + 8R^3 \cdot 4R\sqrt{\frac{2\log(4/\delta)}{n}} \right]$$

$$= \frac{d\,\|\mathbf{M}\|_2^2}{d'} \left[ (9C_3)R^4 \left( \sqrt{\frac{\log(4d/\delta)}{n}} + \frac{\log(4d/\delta)}{n} \right) + 32\sqrt{2}\,R^4\sqrt{\frac{\log(4/\delta)}{n}} \right].$$

Finally, since $\log(4/\delta) \leq \log(4d/\delta)$ for $d \geq 1$, the last term in (64) can be absorbed into the $\sqrt{\log(4d/\delta)/n}$ term. Therefore, there exists an absolute constant $C > 0$ such that, with probability at least $1 - \delta$,

$$\left| V(\mathbf{X}) - V(\widehat{\mathbf{X}}) \right| \leq \frac{d \, \|\mathbf{M}\|_2^2}{d'} \, C \, R^4 \left( \sqrt{\frac{\log(4d/\delta)}{n}} + \frac{\log(4d/\delta)}{n} \right). \tag{64}$$

Absorbing constant factors inside the logarithm (i.e., replacing $\log(4d/\delta)$ by $\log(d/\delta)$) yields the bound stated in Theorem 4.2.

$\square$

# B. Experiment Details and Additional Analysis

## B.1. Experiment Details and Additional Results of Figure 2

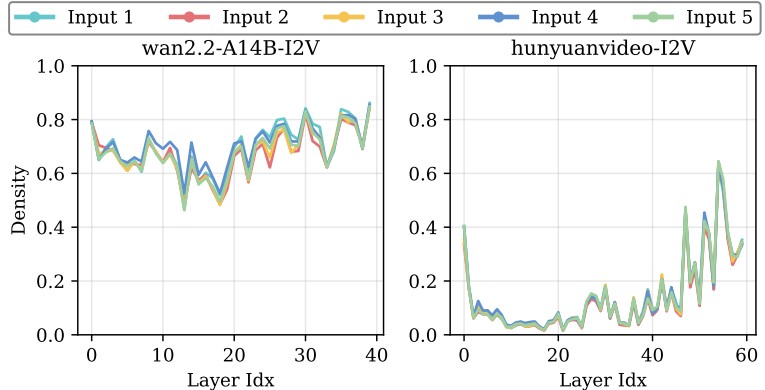

*Figure 10.* Additional layer-wise attention sparsity across different models.

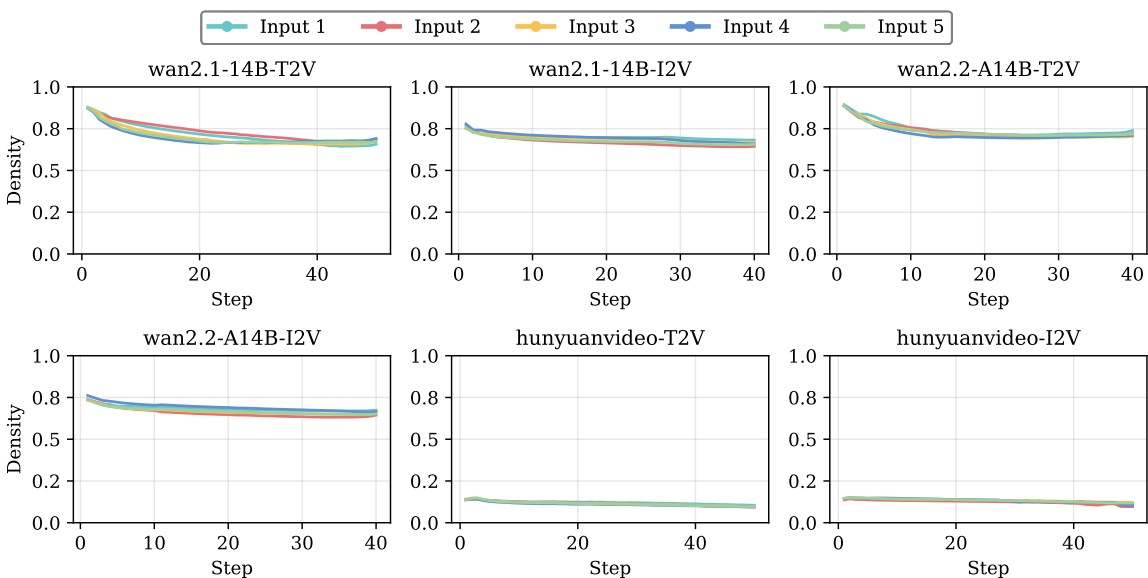

*Figure 11.* Layer attention sparsity across different steps of representative models.

We conduct the experiments in Figure 2 to analyze **layer-wise heterogeneity** and **layer-wise stability** of attention sparsity in video diffusion transformers. Specifically, we randomly sample a small set of inputs from VBench (Huang et al., 2024)

for text-to-video models and from VBench++ (Zheng et al., 2025) for image-to-video models. Experiments are performed at resolutions 720p, corresponding to $720 \times 1280$. For Wan-series models, videos contain 81 frames, while HunyuanVideo-series models use 129 frames. For each layer, we measure the attention density, defined as the minimum fraction of attention entries required to cover 80% of the cumulative attention mass. We evaluate models including Wan2.1-T2V-1.3B, Wan2.1-T2V-14B, Wan2.1-I2V-14B, Wan2.2-T2V-A14B, Wan2.2-I2V-A14B, HunyuanVideo-T2V, and HunyuanVideo-I2V. Figure 2 reports some attention density. Results for the remaining models are provided in Figure 10, and the attention density across different diffusion steps of layers is reported in Figure 11. We observe that the attention sparsity of a specific layer is minimally influenced by varying inputs, suggesting that sparsity is an intrinsic property inherent in the model.

