# OpenReview forum: "Attention Sparsity is Input-Stable: Training-Free Sparse Attention for Video Generation via Offline Sparsity Profiling and Online QK Co-Clustering"
_ICML.cc/2026/Conference — ICML 2026 regular_

### Official Review · Reviewer_BEB3 · 2026-02-26

**Soundness:** 3
**Presentation:** 3
**Significance:** 3
**Originality:** 2
**Overall Recommendation:** 4
**Confidence:** 4

**Summary:**

This paper proposes a training-free framework for accelerating video diffusion transformers by exploiting the inherent sparsity of attention maps. The core idea is to replace dense full-attention with block-sparse attention, where only a subset of query-key (Q-K) block pairs with high attention mass are computed. The authors demonstrate SVOO on multiple video DiT architectures achieving large end-to-end speedups with minimal quality degradation.

**Compliance With Llm Reviewing Policy:**

Affirmed.

**Final Justification:**

The rebuttal substantially addresses several of my concerns. I therefore raise my score.

**Key Questions For Authors:**

1. How sensitive is the offline co-clustering to the choice of calibration prompts? If the calibration set consists of simple prompts (e.g., "a cat walking") but the test set contains complex multi-object scenes, does the block structure still transfer effectively?

2. What is the peak GPU memory usage of SVOO compared to the dense baseline? Sparse attention can sometimes increase memory due to index bookkeeping and irregular memory access patterns. If SVOO reduces memory usage, this would be an additional practical benefit worth highlighting.

3. Are there specific types of video content (e.g., videos with many small moving objects, rapid scene changes, or fine-grained textures) where the block-sparse approximation introduces visible artifacts?

**Limitations:**

The authors does not address several important limitations: (1) the potential sensitivity of offline profiling to calibration data distribution, (2) the lack of automatic sparsity level selection, (3) the absence of failure case analysis, and (4) the unclear interaction between sparse attention and other acceleration techniques.

**Strengths And Weaknesses:**

### Strengths

1. The offline-online decomposition is a well-motivated and principled design that cleanly separates input-invariant structure discovery from input-dependent dynamic selection.

2. The empirical evaluation covers multiple state-of-the-art video DiT architectures, demonstrating broad applicability. The method is tested on four distinct model families: Wan2.1-1.3B, Wan2.1-14B, Wan2.2, and HunyuanVideo, showing consistent improvements.

3. The method achieves meaningful wall-clock speedups without requiring any model retraining or fine-tuning. The training-free nature makes the method immediately deployable on existing pretrained models, lowering the barrier to adoption compared to methods requiring distillation or architectural changes.

### Weaknesses

1. The paper omits comparison with other training-free sparse attention methods such as Sparse-vDiT (arXiv:2506.03065), which also targets video diffusion transformers with sparse attention. ToCa (Token-wise Feature Caching, arXiv:2410.05317) and SpeCa (Speculative Feature Caching, arXiv:2509.11628) are not compared, despite being relevant training-free acceleration methods for diffusion transformers. In addition, the caching baselines (Delta-DiT, TeaCache) are orthogonal to sparse attention and could potentially be combined with SVOO, but no such combination experiments are presented, leaving unclear whether the gains are complementary.

2. The offline profiling phase introduces assumptions and costs that are insufficiently analyzed. Figure 5 shows diminishing returns but does not analyze failure modes or distribution sensitivity; what happens when calibration prompts differ significantly from test prompts in domain or complexity? In addition, the offline profiling must be repeated for each new model architecture, but the paper does not report the wall-clock cost of the offline phase itself, making it difficult to assess the total deployment overhead.

3. The theoretical justification for why spectral co-clustering is the right choice for discovering block-sparse structure is weak. Section 3.2 states that spectral co-clustering is used to partition Q and K tokens, but does not provide theoretical analysis of why this particular clustering method is optimal or even well-suited for attention matrices compared to alternatives (e.g., k-means on token embeddings, hierarchical clustering, or learned partitions).

4. The online phase requires computing block-level mean attention scores at every layer and every denoising step. The paper does not provide a detailed breakdown of the overhead as a fraction of total inference time.  For models with many blocks, this overhead could become non-trivial, but no scaling analysis is provided.


5. The paper does not compare the online selection overhead against simpler heuristics (e.g., fixed sparsity patterns derived purely from offline profiling without online adaptation), making it unclear how much the online component contributes relative to its cost.


6. No comparison is made on inference cost metrics beyond wall-clock time, such as peak GPU memory usage, which is critical for deployment on resource-constrained hardware. The evaluation does not provide statistical significance tests (e.g., confidence intervals, p-values), making it hard to draw reliable conclusions.

---

> ### Author Rebuttal · Authors · 2026-03-29
>
> We sincerely thank the reviewer for detailed and insightful questions!
>
> ---
>
> **Q1: Additional baselines**
>
> **A1:** Thanks!
>
> - For **Sparse-vDiT, we provide the comparison below** with Fig 13 settings, and **will include and cite this baseline in the revised manuscript**
>
>
>     |  |  | Sparse-vDiT | SVOO |
>     | --- | --- | --- | --- |
>     | Wan2.1-1.3B | PSNR | 25.12 | 29.86 |
>     |  | Latency | 223 | 209 |
>     | Wan2.1-14B | PSNR | 26.81 | 28.24 |
>     |  | Latency | 1219 | 1203 |
>
>     As shown, **our SVOO outperforms Sparse-vDiT**
>
> - ToCa and SpeCa, both great works focus on caching, whereas we target sparse attention; **as we focus on different dimensions, a direct comparison is less straightforward**
> - ToCa, SpeCa, $\Delta$-DiT, and TeaCa **are orthogonal to SVOO and can be integrated for further acceleration. Due to rebuttal limits, we select SVOO+$\Delta$-DiT (b=15)** below
>
>
>     | Latency | SVOO | SVOO+$\Delta$-DiT |
>     | --- | --- | --- |
>     | Wan2.1-1.3B | 209 | 143 |
>     | Wan2.1-14B | 1219 | 876 |
> - **All five great works are highly relevant to SVOO, and we will incorporate them into related work** for completeness
>
> ---
>
> **Q2: Sensitivity of the calibration prompts**
>
> **A2**: Thanks!
>
> - **We further report Wan2.1-1.3B results with 10 random VBench calibration prompts**
>
>
>     | PSNR | Latency |
>     | --- | --- |
>     | 29.88 | 209 |
> - We also **report results with different calibration sizes**
>
>
>     | #Calibration | PSNR | Latency |
>     | --- | --- | --- |
>     | 1 | 29.81 | 209 |
>     | 5 | 29.79 | 208 |
>
>     As shown, due to **our key insight that attention sparsity varies little across inputs, SVOO is insensitive to calibration prompts**
>
>
> ---
>
> **Q3: Cost of the offline phase**
>
> **A3:** Thanks!
>
> - The offline stage **needs only 10 inference runs and adds minor cost**
> - Moreover, **offline profiling is a one-time cost amortized over subsequent runs**, making it economical
> - We further report the per-prompt offline runtime
>
>
>     |  | Profiling | Dense |
>     | --- | --- | --- |
>     | Wan2.1-1.3B | 8m 4s | 6m 57s |
>     | Wan2.1-14B | 39m 5s  | 33m 2s |
>
>     As shown, the offline runtime is comparable to dense attention and incurs **a one-time cost with lasting benefits**
>
>
> ---
>
> **Q4: Why is our co-clustering better suited to sparse attention**
>
> **Q4:** Thanks!
>
> - **Intuitively,** **previous independent clustering ignores the bilinear interaction** $A=QK^{\top}$, while our co-clustering avoids this
> - **Theoretically,** the independent **k-means in SVG2 optimizes** $\min \sum_{i} \Vert q_{i} - c_{i}^{q}\Vert + \min \sum_{i} \Vert k_{i} - c_{i}^{k}\Vert$, which is **misaligned with sparse attention**
>
>     In contrast, **our co-clustering optimizes** $\min \sum_{i,j} \Vert q_{i}c_{k}^{j} - c_{i}^{q}C_{k} \Vert - \min \sum_{i} \Vert k_{i}c_{q}^{j} - c_{i}^{k}C_{Q} \Vert$  which is **aligned with sparse attention**, grouping queries and keys with similar attention behavior. Here, $c$ is cluster center
>
> - **Empirically,** we investigate the attention recall
>
>
>     |  | K-means | Ours |
>     | --- | --- | --- |
>     | wan2.1-1.3B | 78.7% | 84.2% |
>     | wan2.1-14B | 88.5% | 90.9% |
>     | wan2.2-14B | 89.9% | 92.5% |
>
>     As shown, **our co-clustering achieves higher attention recall**
>
>
> ---
>
> **Q5: The computational cost of block-level mean attention scores**
>
> **Q5:** Thanks!
>
> - We report the cost of computing mean attention scores
>
>
>     |  | percentage |
>     | --- | --- |
>     | wan2.1-1.3B | 0.47% |
>     | wan2.1-14B | 0.22% |
>
>     It is worth noting that **computing mean attention scores is unavoidable in all block-wise sparse attention** (e.g. SVG2, SpargeAttn), as it is needed to estimate block importance
>
>
> ---
>
> **Q6: The contribution of the online component**
>
> **A6:** Thanks!
>
> - **We provide further ablation results** of the online part
>
>
>     |  | PSNR | Latency |
>     | --- | --- | --- |
>     | all | 29.86 | 209 |
>     | w/o Q Co-Clustering | 29.35 | 221 |
>     | w/o K Co-Clustering | 29.17 | 220 |
>     | w/o QK Co-Clustering | 29.10 | 237 |
>
>     As shown, **our online part are important for both quality and efficiency**
>
>
> ---
>
> **Q7: GPU memory usage of SVOO**
>
> **A7:** Thanks!
>
> - **We report the peak GPU memory**
>
>
>     |  | SVOO | SVG2 |
>     | --- | --- | --- |
>     | Wan2.1-1.3B | 21GB | 23GB |
>     | Wan2.1-14B | 64GB | 67GB |
>
>     As shown, **our SVOO uses lower GPU memory than baselines**
>
>
> ---
>
> **Q8: Performance on challenging cases**
>
> **A8:** Thanks!
>
> - We report results **on challenging VBench cases** covering the mentioned scenarios (idx [262, 661, 694, 697, 701, 710, 719, 724, 728, 737])
>
>
>     |  | SVOO |  | SVG2 |  |
>     | --- | --- | --- | --- | --- |
>     |  | PSNR | Latency | PSNR | Latency |
>     | Wan2.1-1.3B | 28.68 | 212 | 27.91 | 239 |
>     | Wan2.1-14B | 25.42 | 1210 | 25.17 | 1266 |
>
>     As shown, we **achieve better performance on challenging cases**

---

> > ### Author Rebuttal · Reviewer_BEB3 · 2026-04-03
> >
> > Thank you for the response. It addresses some of my concerns. I still hope to see a direct comparison with cache-based acceleration methods in future revisions, as they are also training-free. Overall, my view of the paper is somewhat positive, but not enough to warrant an increase in score.

---

> > > ### Author Response · Authors · 2026-04-04
> > >
> > > Thank you very much for your careful review and suggestions!
> > >
> > > - **[Directly Compare with Cache-Based Method]**
> > >
> > >     To address your concern, we **conduct a direct comparison between our SVOO and representative cache-based acceleration approaches** with the same settings (Fig 13, Warm-up=20%, 81 frames, 720P)**.**
> > >
> > >     Specifically, we include ToCa[1] and SpeCa[2] in our evaluation.The hyperparameters for ToCa are $\mathcal{N}=2$,$\mathcal{R}=70%$ , and for SpeCa are $\tau_0=0.1,K=3$ (Notions follow original papers)
> > >
> > >     |  | Wan2.1-1.3B-T2V |  | Wan2.1-14B-T2V |  |
> > >     | --- | --- | --- | --- | --- |
> > >     |  | PSNR(dB)$\uparrow$ | Latency(s)$\downarrow$ | PSNR(dB)$\uparrow$ | Latency(s)$\downarrow$ |
> > >     | Dense | - | 419 | - | 1982 |
> > >     | ToCa(Cache-based) | 26.38 | 239 | 25.62 | 1252 |
> > >     | SpeCa(Cache-based) | 29.33 | 220 | OOM | OOM |
> > >     | **SVOO(Ours)** | **29.86** | **209** | **28.24** | **1203** |
> > >
> > >     As shown, **our SVOO consistently achieves the best PSNR while maintaining the lowest latency**
> > >
> > > - **[More Comprehensive Comparison]**
> > >
> > >     **To ensure a more comprehensive comparison, we further evaluate SVOO against ToCa and SpeCa across a range of hyperparameters.**
> > >
> > >     |  |  | Wan2.1-1.3B-T2V |  | Wan2.1-14B-T2V |  |
> > >     | --- | --- | --- | --- | --- | --- |
> > >     | Method | ToCa Config | PSNR(dB)$\uparrow$ | Latency(s)$\downarrow$ | PSNR(dB)$\uparrow$ | Latency(s)$\downarrow$ |
> > >     | ToCa | $\mathcal{N}=2$ | 26.38 | 239 | 25.62 | 1252 |
> > >     |  | $\mathcal{N}=3$ | 25.96 | 196 | 24.73 | 1077 |
> > >     |  | $\mathcal{N}=4$ | 24.46 | 164 | 24.42 | 912 |
> > >     |  | $\mathcal{N}=5$ | 24.19 | 149 | 23.24 | 838 |
> > >     |  | $\mathcal{N}=6$ | 24.02 | 137 | 23.12 | 802 |
> > >     | **SVOO (Ours)** | - | **29.86** | 209 | **28.24** | 1203 |
> > >
> > >     |  |  | Wan2.1-1.3B-T2V |  | Wan2.1-14B-T2V |  |
> > >     | --- | --- | --- | --- | --- | --- |
> > >     |  | SpeCa Config | PSNR(dB)$\uparrow$ | Latency(s)$\downarrow$ | PSNR(dB)$\uparrow$ | Latency(s)$\downarrow$ |
> > >     | SpeCa | $\tau_{0}=0.05$ | 29.78 | 224 | OOM | OOM |
> > >     |  | $\tau_{0}=0.10$ | 29.33 | 220 | OOM | OOM |
> > >     |  | $\tau_{0}=0.30$ | 28.84 | 210 | OOM | OOM |
> > >     |  | $\tau_{0}=0.50$ | 28.78 | 204 | OOM | OOM |
> > >     |  | $\tau_{0}=1.00$ | 27.97 | 197 | OOM | OOM |
> > >     | **SVOO (Ours)** | - | **29.86** | 209 | **28.24** | 1203 |
> > >
> > >     As shown, **SVOO outperforms both ToCa and SpeCa in quality across all settings (up to +5.84 dB over ToCa)**, demonstrating a superior quality-efficiency tradeoff. Notably, SpeCa fails on the Wan2.1-14B-T2V model due to OOM (H200), whereas SVOO runs successfully
> > >
> > > - **[GPU Memory Consumption Comparison]**
> > >
> > >     To better understand the practical advantages of SVOO, **we report GPU memory consumption** across methods:
> > >
> > >     |  | Wan2.1-1.3B-T2V | Wan2.1-14B-T2V |
> > >     | --- | --- | --- |
> > >     | ToCa | 41GB | 137GB |
> > >     | SpeCa | 99GB | ≥141GB (OOM) |
> > >     | **SVOO(Ours)** | **21GB** | **64GB** |
> > >     | **SVOO Improve Over ToCa** | **20 GB $\uparrow$ (41GB$\rightarrow$21GB)** | **73 GB $\uparrow$ (137GB$\rightarrow$64GB)** |
> > >     | **SVOO Improve Over SpeCa** | **78 GB $\uparrow$ (99GB$\rightarrow$21GB)** | **≥ 77 GB $\uparrow$ (≥141GB$\rightarrow$64GB)** |
> > >
> > >     As shown, **SVOO reduces GPU memory by up to 78 GB over SpeCa and 73 GB over ToCa**, making it far more suitable for resource-constrained deployment.
> > >
> > > - **[Orthogonal Integration]**
> > >
> > >     Moreover, SVOO is **highly orthogonal** to cache-based acceleration methods and can be **seamlessly integrated for more aggressive acceleration**. Results on Wan2.1-1.3B-T2V:
> > >
> > >     |  | PSNR(dB)$\uparrow$ | Latency(s)$\downarrow$ |
> > >     | --- | --- | --- |
> > >     | SVOO | 29.86 | 209 |
> > >     | SVOO+ToCa | 23.39 | 171 |
> > >     | SVOO+SpeCa | 24.31 | 159 |
> > >
> > >     As shown, **combining SVOO with cache-based methods yields further latency reductions**, confirming the **two paradigms are complementary**
> > >
> > >
> > > ---
> > >
> > > ## **In summary**
> > >
> > > ***(1) [Better Generation Quality]:***  Our sparse attention method SVOO achieves **comparable or faster acceleration** than cache-based methods, **while delivering superior video quality**.
> > >
> > > ***(2) [Lower GPU Memory Overhead]:*** Our SVOO **consumes significantly less GPU memory** **(up to 78 GB less)**, making it more suitable for resource-constrained scenarios.
> > >
> > > ***(3) [Orthogonal Compatibility]:*** **Our sparse attention acceleration method SVOO is orthogonal to cache-based methods** and can be seamlessly integrated for further acceleration, offering a flexible and composable framework.
> > >
> > > ---
> > >
> > > We hope our response will address your concerns. We will incorporate these direct comparisons with cache-based acceleration methods into the revised manuscript.
> > >
> > > Thank you again for your detailed and insightful comments!
> > >
> > > ---
> > >
> > > *[1] Accelerating diffusion transformers with token-wise feature caching. ICLR 25*
> > >
> > > *[2] SpeCa: Accelerating Diffusion Transformers with Speculative Feature Caching. ACM MM 25*

---

### Official Review · Reviewer_esyC · 2026-03-09

**Soundness:** 4
**Presentation:** 4
**Significance:** 3
**Originality:** 3
**Overall Recommendation:** 5
**Confidence:** 3

**Summary:**

The paper proposes a training-free method to convert dense attention video diffusion transformer into a sparse attention one for inference acceleration. Compared to prior work, the proposed method SVOO applies different sparsity threshold on different layers, where the threshold is offline calibrated on a few example prompts prior to inference. Additionally, SVOO applies query-dependent key clustering, where the clustering is computed online but cached for every N steps. The experiment shows that SVOO outcompetes prior training-free methods on Wan and Hunyuan models by obtaining better VBench performance while having the biggest acceleration.

**Compliance With Llm Reviewing Policy:**

Affirmed.

**Final Justification:**

The rebuttal have addressed my concerns. I will maintain my score.

**Key Questions For Authors:**

1. Paper mentioned the use of custom triton kernel but does not specify the speed up compared to native implementation baseline. Is it necessary?
2. How does the method compare against training-based methods?
3. How do you expect it to work on step-distilled models?

**Limitations:**

1. The method requires more "finetuning" compared to prior work. It requires pre-computing the layer-wise sparsity pattern prior to inference. This part of the overhead may not be suitable for some applications that only runs the model a few time.
2. The comparisons are against other training-free methods. It is unknown how it performance against training-based mehods, and also how it performs on step-distilled models. There are open-source step-distilled models, which can be used to study for future work.

**Strengths And Weaknesses:**

1. The method is intuitive and sound. The two findings, regarding to the layer-wise sparsity tuning and the query-dependent key clustering, is clearly motivated.
2. Good presentation on the motivation and the proposed methodology. The experimentation also shows strong results against prior method.
3. Method is original. The contributions are clearly distinguished.

---

> ### Author Rebuttal · Authors · 2026-03-29
>
> We sincerely thank the reviewer for the detailed comments and insightful questions! Responses are as follows.
>
> **Q1: The contribution of the custom Triton kernel.**
>
> **A1:** Thanks!
>
> - We provide the ablation results without our custom Triton kernel as follows:
>
>
>     |  | Latency w/ Triton | Latency w/o Triton | Time saved | Reduction |
>     | --- | --- | --- | --- | --- |
>     | wan2.1-1.3B-T2V | 209s | 254s | 45s | 17.7% |
>     | wan2.1-14B-T2V | 1210s | 1477s | 267s | 18.1% |
>     | wan2.2-14B-T2V | 981s | 1215s | 234s | 19.3% |
>     | HunyuanVideo-T2V | 832s | 1163s | 331s | 28.5% |
>
>     As we can see, **our custom Triton kernel plays a crucial role in improving efficiency.**
>
>
> ---
>
> **Q2: How does the method compare against training-based methods?**
>
> **A2:** Thanks!
>
> - Training-based and training-free methods **each have their own advantages**. We report the comparison with VSA[1] below on Wan2.1-14B-T2V using their official Hugging Face weights:
>
>
>     |  | SVOO | VSA |
>     | --- | --- | --- |
>     | ImageQual $\uparrow$ | **68.92%** | 68.07% |
>     | AesQual $\uparrow$ | **61.01%** | 58.25% |
>     | SubConsist $\uparrow$ | **97.67%** | 97.65% |
>     | BackConsist $\uparrow$ | **97.69%** | 97.34% |
>     | Training Cost $\downarrow$ | **0 (Totally free)** | **32 H200 * 14 hours** |
> - As shown, while **training-based methods** can often achieve more aggressive acceleration, they **require additional training overhead.**  In contrast, **training-free sparse attention is more suitable for resource-constrained scenarios, as it does not require any training cost.**
>
> ---
>
> **Q3: Cost of the offline phase**
>
> **A3:** Thanks!
>
> - The offline stage **needs only 10 inference runs and adds minor cost**
> - Moreover, **our offline profiling is a one-time cost amortized over subsequent runs**, making it economical in practice
> - We further report the per-prompt offline runtime
>
>
>     |  | Profiling | Dense |
>     | --- | --- | --- |
>     | Wan2.1-1.3B | 8m 4s | 6m 57s |
>     | Wan2.1-14B | 39m 5s  | 33m 2s |
>
>     As shown, the offline runtime is comparable to dense attention and incurs **a one-time cost with lasting benefits**
>
> - Additionally, we **analyze the sensitivity to the number of calibration prompts** used in the offline stage, with different prompt counts randomly selected from VBench. The results are as follows:
>
>
>     | #Calibration Prompts | PSNR | Latency |
>     | --- | --- | --- |
>     | 1 | 29.81 | 209 |
>     | 5 | 29.79 | 208 |
>     | 10 | 29.88 | 209 |
>
>     As shown, due to our key insight that attention sparsity varies little across inputs, **SVOO is largely insensitive to the choice and number of calibration prompts**, and **even a much smaller calibration set still yields strong performance, making the offline stage feasible and economical in practice.**
>
>
> ---
>
> **Q4: How is SVOO expected to perform on step-distilled models?**
>
> **A4**: Thanks!
>
> - Due to our co-clustering results remaining relatively stable across different diffusion steps, we do not **need to perform clustering at each diffusion step**. In addition, SVOO requires only 2 iterations for its initial co-clustering, which introduces minimal overhead. **These properties suggest that SVOO can still provide meaningful efficiency gains on step-distilled models.**
>
> ---
>
> *[1] Faster Video Diffusion with Trainable Sparse Attention, NeurIPS 2025*

---

> > ### Author Rebuttal · Reviewer_esyC · 2026-03-31
> >
> > Rebuttal resolves my question

---

> > > ### Author Response · Authors · 2026-04-01
> > >
> > > We sincerely thank the reviewer for the careful reading of our paper and for the constructive and insightful feedback. We appreciate your recognition of our work!
> > >
> > > We are grateful for your thoughtful questions, which helped us further clarify and improve the paper. We are glad that our response has addressed your concerns!
> > >
> > > **Thank you again for your time and valuable suggestions!**

---

### Official Review · Reviewer_B6c4 · 2026-03-11

**Soundness:** 2
**Presentation:** 3
**Significance:** 3
**Originality:** 2
**Overall Recommendation:** 4
**Confidence:** 4

**Summary:**

This paper studies how to accelerate DiT video generation using training-free sparse attention. The authors argue that existing block-wise sparse attention methods overlook two factors: layer-wise heterogeneity in attention sparsity and the coupling between query and key representations when constructing block partitions. To address this, they propose SVOO, a framework that combines offline layer-wise sparsity profiling with an online bidirectional co-clustering algorithm. The offline stage estimates a sparsity schedule for each transformer layer based on attention statistics from a calibration set. The online stage clusters query and key tokens to form better-aligned attention blocks and selects salient block pairs for computation. Experiments on several recent video diffusion models across text-to-video and image-to-video tasks show that the method achieves improved quality–efficiency trade-offs compared with prior training-free sparse attention baselines.

**Compliance With Llm Reviewing Policy:**

Affirmed.

**Final Justification:**

The rebuttal addressed most of my concerns. I think the authors should add the experiments to the revised paper. The presentation also needs improving because some typos are found in the original submission. Therefore I want to maintain my acceptance.

**Key Questions For Authors:**

1. How much impact does the proposed thresholding mechanism actually have (W2)
2. Attention density alone may not capture this timestep heterogeneity, potentially leading to similar sparsity settings across non–warm-up steps despite their different impact on final quality. (W3)
3. SVOO also constructs blocks via clustering, does the method perform a similar permutation, and what overhead does this introduce in the inference pipeline? (MW1)

**Limitations:**

yes

**Strengths And Weaknesses:**

**Strengthes**
1. **Timely and significant problem.** The paper addresses the increasingly important problem of accelerating diffusion transformer–based video generation, which is computationally expensive due to dense spatio-temporal attention. Improving inference efficiency for large video generation models is a timely and practically significant research direction.
2. **Intuitive and interesting algorithmic idea.** The proposed bidirectional co-clustering mechanism for jointly partitioning queries and keys is a novel design choice compared with prior methods that cluster them independently. The idea is intuitive in that attention patterns are determined by query–key interactions, and jointly modeling them may lead to better-aligned sparse attention blocks.
3. **Practical and deployable approach.** The method is training-free and can be applied to existing video diffusion models without retraining or modifying model parameters, which makes it potentially practical for real-world deployment and integration into existing inference pipelines.

**Weaknesses**
1. Potentially misleading characterization of prior work. The paper claims that prior training-free sparse attention methods typically apply uniform sparsity ratios across layers. However, several mentioned prior works (e.g., SpargeAttention and SVG2) already employ adaptive selection strategies such as topcdf / top-p block selection, where the effective compute budget varies across layers, heads, and inputs depending on the attention distribution. In this sense, the proposed layer-wise sparsity schedule appears closer to introducing an upper bound (cap) on the sparsity level rather than fundamentally enabling dynamic sparsity.
2. The sparsity cap is unlikely to be triggered in practice. The block selection rule introduces a threshold \theta = 0.1 to determine whether the sparsity schedule acts as a lower or upper bound. However, the estimated sparsity budgets s derived from offline profiling are typically much larger than 0.1 in sparse attention settings (often above 0.5). As a result, the branch s <= \theta  may rarely be triggered in practice, and the rule effectively degenerates to selecting. This raises questions about whether the proposed thresholding mechanism meaningfully affects the behavior of the method.
3. The calibration metric may not correlate well with generation quality. The offline sparsity schedule is derived from attention density statistics, assuming that layers with denser attention are more sensitive to pruning. However, in diffusion models, generation quality is strongly dependent on the diffusion timestep. Early timesteps often have a larger impact on the final generation quality even if their attention patterns appear similarly sparse. Therefore, using attention density alone as the calibration signal may not accurately reflect the true importance of different layers or steps, potentially leading to overly uniform sparsity schedules across non–warm-up steps.

**Minor Weaknesses**
1. Prior work such as SVG2 explicitly permutes tokens belonging to the same cluster into contiguous memory layouts. SVOO also relies on clustering-based block construction, it likely requires a similar permutation. However, the paper does not clearly describe whether such a layout transformation is performed. Clarifying this aspect would help improve reproducibility and better explain the reported efficiency gains.
2. Typos: “Traning-free Sparse Attenion” in Section 2.2.

---

> ### Author Rebuttal · Authors · 2026-03-29
>
> We sincerely thank the reviewer for the detailed comments and insightful questions! Responses are as follows.
>
> ---
>
> **Q1: The effect of the threshold parameter $\theta$.**
>
> **A1:** Thanks!
>
> - We provide the experimental results using different values of the threshold $\theta$ of our SVOO and SVG2 as follows:
>
>
>     |  | SVOO |  | SVG2 |  |
>     | --- | --- | --- | --- | --- |
>     | $\theta$ | PSNR | Latency | PSNR | Latency |
>     | 0.00 | **26.75** | **195** | 22.53 | 219 |
>     | 0.05 | **28.24** | **197** | 26.41 | 225 |
>     | 0.10 | **29.86** | **209** | 29.18 | 239 |
>     | 0.15 | **30.29** | **213** | 29.74 | 247 |
>     | 0.20 | **30.69** | **223** | 30.22 | 258 |
>
>     As shown, **SVOO still performs well even when $\theta = 0$, whereas SVG2 suffers a significant performance drop.** This suggests that the **offline stage helps push the sparsity level closer to a safe upper bound.**
>
>
> ---
>
> **Q2: The relationship between attention sparsity and pruning sensitivity, given the timestep-dependent quality importance in diffusion models.**
>
> **A2**: Thanks!
>
> - We **also observe that early diffusion timesteps have a greater impact on the final generation quality**. However, at the same time, we also find that **the attention sparsity (fraction of entries for 90% attention mass) in the early diffusion timesteps is denser**, as shown below:
>
>
>     | Attention density (lower indicates higher sparsity) | Wan2.1-1.3B-T2V | Wan2.1-14B-T2V |
>     | --- | --- | --- |
>     | Step 5 | 0.7330 | 0.7922 |
>     | Step 10 | 0.6284 ($\downarrow$ 0.1046) | 0.7461 ($\downarrow$ 0.0461) |
>     | Step 15 | 0.5644 ($\downarrow$ 0.0640) | 0.7183 ($\downarrow$ 0.0278) |
>     | Step 20 | 0.5285 ($\downarrow$ 0.0359) | 0.6963 ($\downarrow$ 0.0220) |
>     | Step 25 | 0.4997 ($\downarrow$ 0.0288) | 0.6852 ($\downarrow$ 0.0110) |
>     | Step 30 | 0.4709 ($\downarrow$ 0.0288) | 0.6781 ($\downarrow$ 0.0071) |
>     | Step 35 | 0.4430 ($\downarrow$ 0.0279) | 0.6719 ($\downarrow$ 0.0062) |
>     | Step 40 | 0.4208 ($\downarrow$ 0.0222) | 0.6658 ($\downarrow$ 0.0061) |
>     | Step 45 | 0.4032 ($\downarrow$ 0.0176) | 0.6643 ($\downarrow$ 0.0015) |
>     | Step 50 | 0.4159 ($\uparrow$ 0.0127) | 0.6770 ($\uparrow$ 0.0127) |
>
>     As shown in the table, this trend suggests that **timestep-wise attention sparsity is consistent with pruning sensitivity: early, quality-critical timesteps are less sparse and thus less amenable to pruning**, whereas later timesteps are sparser and more pruning-tolerant.
>
>     **Thus, attention sparsity can still effectively reflect timestep-wise pruning sensitivity.**
>
>
> ---
>
> **Q3: The permutation operation usage in SVOO.**
>
> A3: Thanks!
>
> - Similar to prior clustering-based methods SVG2, **our implementation also performs the corresponding token permutation** transformation for block construction for better quality. **We will clarify this detail in the revised manuscript to improve completeness**.
> - We provide **the cost of the permutation operation** as follows:
>
>
>     |  | Permutation Latency | Percentage vs attention |
>     | --- | --- | --- |
>     | Wan2.1-1.3B-T2V | 2.96s | 1.95% |
>     | Wan2.1-14B-T2V | 9.65 | 1.12% |
>     | Wan2.1-14B-I2V | 7.67 | 0.81% |
>     | Wan2.2-14B-T2V | 1.00 | 0.31% |
>     | Wan2.2-14B-I2V | 0.53 | 0.19% |
>     | HunyuanVideo-T2V | 7.71 | 1.23% |
>     | HunyuanVideo-I2V | 7.72 | 1.20% |
>
>     Here, Permutation Latency includes both tensor permutation by clustering labels (forward rearrangement) and the inverse permutation (backward recovery)**.** As shown, **the permutation incurs a low cost compared to the overall attention computation.**
>
> - **We further report the quality without permutation.** Since our clustering relies on permutation, we remove both permutation and co-clustering, **replacing them with mean pooling using the same number of Q&K blocks.** Results under the same settings as Figure 13 are shown below.
>
>
>     | PSNR | w/ permutation | w/o permutation |
>     | --- | --- | --- |
>     | Wan2.1-1.3B-T2V | 29.86 | 24.37 |
>     | Wan2.1-14B-T2V | 28.24 | 22.52 |
>     | Wan2.1-14B-I2V | 27.06 | 20.93 |
>     | Wan2.2-14B-T2V | 24.85 | 18.98 |
>     | Wan2.2-14B-I2V | 29.13 | 22.81 |
>     | HunyuanVideo-T2V | 24.48 | 22.07 |
>     | HunyuanVideo-I2V | 24.95 | 22.11 |
>
>     Thus, **the permutation overhead is small while yielding better quality**.
>
>
> ---
>
> **Q4: Typo error.**
>
> **A4:** Thanks!
>
> - We **sincerely thank you for pointing this out carefully! We will correct the typo error** in Section 2.2, changing “Traning-free Sparse Attenion” to “Training-free Sparse Attention.”

---

> > ### Author Rebuttal · Reviewer_B6c4 · 2026-04-04
> >
> > It will be better if you add some clarification of W1 in the revised paper.

---

> > > ### Author Response · Authors · 2026-04-04
> > >
> > > Thank you again for your careful review and constructive feedback. Your comments have been very helpful in improving the quality and clarity of our paper. **We will add further clarification on W1 and revise the manuscript accordingly in the revised version as follows:**
> > >
> > > - **Clarification on layer-wise sparsity:**  We will clarify that some prior works already adopt adaptive selection strategies, such as top-p block selection. (e.g. SVG2, SpargeAttn. **Both of them are our important baseline and are already included in our experiments**.) In this sense, our proposed layer-wise sparsity schedule is better understood not simply as introducing dynamic sparsity, but rather as estimating a layer-specific upper bound on the safe sparsity level by profiling each layer’s intrinsic pruning tolerance.
> > > - **Our core contribution 1: We identify an important empirical phenomenon: the attention sparsity of a given layer is only minimally affected by different inputs.** This layer-wise stability allows us to reliably profile sparsity using only a small calibration set.
> > > - **Our core contribution 2: We further provide a theoretical analysis** showing the stability of attention sparsity, **and derive an upper bound** that relates this stability to the token representation radius and the number of tokens.
> > >
> > > **If there are any remaining concerns, we would be very happy to address them at any time!**
> > >
> > > Thank you again for your time and valuable suggestions!

---

### Official Review · Reviewer_FKq6 · 2026-03-12

**Soundness:** 2
**Presentation:** 2
**Significance:** 3
**Originality:** 3
**Overall Recommendation:** 4
**Confidence:** 4

**Summary:**

This paper proposes SVOO, a novel training-free sparse attention framework to address two limitations: layer-wise homogeneity sparsity optimization and the misalignment caused by independent Q-K clustering.
The sparse attention framework adopted in this paper is similar to SVG2 (including QK clustering and block selection operations), with two major optimizations: an additional offline stage for layer-dependent optimization and Q-K coupled clustering.
Extensive experiments are conducted on seven state-of-the-art mainstream video generation models and demonstrate that the proposed method achieves significant inference acceleration while maintaining high visual quality.

**Compliance With Llm Reviewing Policy:**

Affirmed.

**Ethical Review Flag:**

Flag this paper for an ethics review.

**Final Justification:**

The rebuttal have addressed most of my concerns. Although the absolute benefit of the offline stage may appear modest for certain models, the authors have demonstrated its theoretical and practical value in approaching the safe sparsity upper bound. I recommend that the authors include more detailed ablation experiments for both the offline and online stages in the revised version.

**Key Questions For Authors:**

1. Can you provide a correlation analysis between pre-softmax logit variance, attention density, and sparsity ratio to support the layer-wise heterogeneity hypothesis?
2. In the offline stage, why were "10 calibration inputs randomly generated by GPT-5" used? How did you control for the distribution, length, and style of these prompts? Would the results remain robust if a random subset from the VBench benchmark were used instead?
3. What is the theoretical rationale for the parameter selection in Eq. (7)? Additionally, please provide a sensitivity analysis for the parameter $\theta$?
4. The current ablation study only removes either the offline or online module. Could you provide more detailed ablation experiments to isolate the specific contribution fo each component?

**Limitations:**

1. Although the method is conceptually clear, the theoretical basis for offline profiling is relatively weak and relies more on intuitive. Furthermore, the current ablation experiments struggle to support its effectiveness.
2. There are some concerns regarding the fairness of the experiments, especially the warm-up strategy for different backbones.
3. The study lacks more detailed ablation experiments.

**Strengths And Weaknesses:**

Strengths
1. **Insightful Motivation**: The paper analyzes limitations of existing methods. It points out that layer-wise heterogeneity stability and Q-K decoupled blocking may be critical issues in current sparse attention methods.
2. **Extensive and Up-to-date Evaluation**: The authors evaluate SVOO with 7 latest and representative large video generation models (e.g., Wan2.1/2.2, HunyuanVideo, covering T2V and I2V), and the baseline methods (SVG1, SVG2, Radial, SpargeAttn) are up-to-date in this field. The paper reports stable performance improvements of SVOO compared to baseline methods.

Weakness
1. The theoretical analysis in Section 4.1 lacks sufficient validity, and there is noticeable terminology inconsistency.
    - Line 209 claims to study "pre-softmax logits variance," but follows by $a_i = \text{softmax}(z_i(X))$. Line 237 defines $d^k_{l,h}$, while Eq(5) uese $d_{l,h}^{(j)}$
    - The paper uses the row variance of pre-softmax logits as a proxy for attention sparsity. However, the sharpness of the post-softmax distribution depends on multiple factors beyond variance. Relying solely on variance is too simplistic and potentially misleading, especially across different layers. The authors should at least rovide an empirical correlation analysis between logit variance, final attention density, and the tolerable pruning ratio.
2. **Offline Stage**:
    - The conclusion in 4.2 relies on the assumption from 4.1 that "token representations bounded in an R-ball," which is quite broad. Moreover, the assumption $R<<n$ in line 226 lacks theoretical support. If the actual value of $R$ is very large, the upper bound in Eq. (4) (related to $R^4$) may become loose, thereby weakening the hypothesis regarding layer-wise sparsity stability.
    - Experimental results presented in Figure 5 shows that the 'without Off' configuration has almost no difference in quality scores compared to 'Full', and the latency difference between 'Full' and 'without Off' is within 0.03x. This seems to suggest limitations in the practical application of the current layer-wise dynamic sparsity strategy.
3. **Lack of fairer comparison**. Currently, each method selects on point, possibly the optimal point after individual tuning. For example, regarding warm-up, the paper only mentions 'unified warm-up'. It is unclear whether all methods were run under the same warm-up strategy, and the authors do not explain why different warm-up percentage strategies were adopted for different models. This could significantly impact the quality-latency curve. A fairer comparison would be to provide the *warm-up* to *PSNR/latency* curves for other baseline methods as well.

---

> ### Author Rebuttal · Authors · 2026-03-29
>
> We sincerely thank the reviewer for detailed comments and insightful questions!
>
> ---
>
> **Q1: The inconsistent terminology**
>
> **A1:** Thanks!
>
> - We study the **pre-softmax logits variance** because **softmax does not change the relative magnitudes** of attentions.
> - **We will revise the manuscript** to remove the expression $a_{i}=\mathrm{softmax}(\mathbf{z_i}(\mathbf{X}))$ in line 210 and **unify the notation** from $d_{j,h}^{(j)}$ to $d_{j,h}^{k}$ to avoid confusion
>
> ---
>
> **Q2: The reason for using variance as the sparsity proxy.**
>
> **A2:** Thanks!
>
> - The pre-softmax logits variance is a good proxy for sparsity, as **larger pre-softmax variance tends to lead to a sharper post-softmax distribution**. Since the softmax outputs sum to one, **a sharper distribution corresponds to a sparser attention**
> - We further provide an **empirical correlation analysis between pre-softmax variance and final attention density** (entry fraction for 80% mass, measures **sparsity ratio**) on wan2.1-1.3B-T2V
>
>
>     | Pre-softmax variance | 7.90 | 9.20 | 12.56 | 36.33 | 125.50 |
>     | --- | --- | --- | --- | --- | --- |
>     | Attention density | 0.49 | 0.42 | 0.36 | 0.33 | 0.13 |
>
>     As shown, **larger pre-softmax variance tends to produce sparser final attention,** validating it as a sparsity proxy
>
>
> ---
>
> **Q3: Why are token representations bounded in an R-ball practical?**
>
> **A3:** Thanks!
>
> - In Transformers, **normalizations (Layer/RMS Norm) bound feature magnitudes.** Meanwhile, the **feature dimension $d$ (often hundreds) is much smaller than the number of tokens $n$ (often tens of thousands)**. Thus, we typically have $R \ll n$.
> - **We further provide empirical evidence**, conducted on 10 Vbench inputs.
>
>
>     |  | Wan2.1-1.3B-T2V ($n$=75600) | HunyuanVideo-T2V ($n$=119056) |
>     | --- | --- | --- |
>     | Layer 5 | $R_{q}$=25.5, $R_{k}$=24.5 | $R_{q}$=20.1, $R_{k}$=19.5 |
>     | Layer 10 | $R_{q}$=29.5, $R_{k}$=24.0 | $R_{q}$=20.5, $R_{k}$=20.4 |
>     | Layer 20 | $R_{q}$=27.1,  $R_{k}$=27.7 | $R_{q}$=22.9,  $R_{k}$=25.2 |
>
>     **As shown, $R \ll n$ holds in practice.** Here, $R_{q}$ and $R_{k}$ are the maximum radius of queries and keys, respectively.
>
> - What is more, it is **worth noting that a key novel contribution of our work is deriving a theoretical upper bound on attention sparsity**, linked to the boundedness of token representations
>
> ---
>
> **Q4: Benefit of the layer-wise offline strategy**
>
> **A4:** Thanks!
>
> - From Fig. 5, the offline stage **still speeds up Wan2.2 by 17s without quality loss**, suggesting that it brings sparsity closer to its safe upper bound.
>
> ---
>
> **Q5: The warmup settings**
>
> **A5:** Thanks!
>
> - In all our experiments, we **use the same warmup settings for both SVOO and all baselines**. For HunyuanVideo, we adopt fewer warm-up steps, as we observe that it can tolerate higher attention sparsity (in Fig 2)
> - We **further provide PSNR–latency results of the strongest baseline (SVG2) under different warm-up settings** on Wan2.1-1.3B-T2V, with the same settings in Figure 13
>
>
>     |  | PSNR |  | Latency |  |
>     | --- | --- | --- | --- | --- |
>     | Warmups | SVOO | SVG2 | SVOO | SVG2 |
>     | 0% | 20.31 | 18.93 | 190 | 209 |
>     | 10% | 26.49 | 25.78 | 197 | 212 |
>     | 20% | 29.86 | 29.06 | 209 | 237 |
>
>     As shown, **our SVOO outperforms SVG2 across all warmup settings**
>
>
> ---
>
> **Q6: How sensitive is the choice of the GPT-5-generated calibration prompts?**
>
> **A6**: Thanks!
>
> - Our prompt to GPT-5 for T2V is as follows:
>
>     ```jsx
>     Please generate 10 diverse text-to-video prompts for creating 10 different, visually appealing videos.
>     ```
>
> - **We further provide results on Wan2.1-1.3B-T2V using calibration prompts randomly selected from VBench**, with the same settings in Figure 13
>
>
>     | #Calibration Prompts | PSNR | Latency |
>     | --- | --- | --- |
>     | 1 | 29.81 | 209 |
>     | 5 | 29.79 | 208 |
>     | 10 | 29.88 | 209 |
>
>     As shown, **the performance of our SVOO is robust to calibration prompts**
>
>
> ---
>
> **Q7: Sensitivity analysis for the parameter $\theta$**
>
> **A7**: Thanks!
>
> - We **provide the sensitivity** of $\theta$ on Wan2.1-1.3B-T2V:
>
>
>     | $\theta$ | PSNR | Latency |
>     | --- | --- | --- |
>     | 0.00 | 26.75 | 195 |
>     | 0.05 | 28.24 | 197 |
>     | 0.10 | 29.86 | 209 |
>     | 0.15 | 30.29 | 213 |
>     | 0.20 | 30.69 | 223 |
>
>     As shown, **our SVOO performs well across a wide range of $\theta$**
>
>
> ---
>
> **Q8: Further ablation experiments.**
>
> A8: Thanks! We **provide further ablation experiments beyond Fig 5**
>
> - w/o Q Co-Clustering: replace Q partitioning with k-means, keep K unchanged
> - w/o K Co-Clustering: replace K partitioning with k-means, keep Q unchanged
>
>
>     |  | PSNR | Latency |
>     | --- | --- | --- |
>     | all | 29.86 | 209 |
>     | w/o Q Co-Clustering | 29.35 | 221 |
>     | w/o K Co-Clustering | 29.17 | 220 |
>
>     As shown, **each component contributes to the performance**

---

> > ### Author Rebuttal · Reviewer_FKq6 · 2026-04-03
> >
> > The authors have addressed most of my concerns. Although the absolute benefit of the offline stage may appear modest for certain models, the authors have demonstrated its theoretical and practical value in approaching the safe sparsity upper bound. I recommend that the authors include more detailed ablation experiments for both the offline and online stages in the revised version.

---

> > > ### Author Response · Authors · 2026-04-03
> > >
> > > Thank you very much for your careful review and constructive feedback! **We sincerely appreciate your insightful comments, which have significantly helped improve the quality and clarity of our paper!**
> > >
> > > In particular, we are grateful for your suggestion to provide more detailed ablation studies for both the offline and online stages. We fully agree that such analyses would strengthen the paper. **In the revised manuscript, we will incorporate the additional ablation experiments as presented in the rebuttal to better isolate and quantify the contribution of each component.**
> > >
> > > In summary, our ablation of more detailed ablation experiments for both the offline and online stages with the same settings as Figure 13 on Wan2.1-1.3B-T2V is as follows:
> > >
> > > |  | PSNR | Latency |
> > > | --- | --- | --- |
> > > | all | 29.86 | 209 |
> > > | w/o Q Co-Clustering (Online) | 29.35 | 221 |
> > > | w/o K Co-Clustering (Online) | 29.17 | 220 |
> > > | w/o QK Co-Clustering (Online) | 29.10 | 237 |
> > > | w/o Offline  | 29.77 | 218 |
> > >
> > > **We will incorporate more ablation results into the manuscript to further improve clarity and completeness.**
> > >
> > > Thank you again for your valuable feedback and support!

---

### Decision · Program_Chairs · 2026-04-30

**Decision:**

Accept (regular)

**Comment:**

This paper studies an important and timely problem, namely training-free acceleration of video diffusion transformers through sparse attention. Reviewers generally agree that the paper is practically meaningful, well motivated, and empirically strong, especially given its broad evaluation across multiple recent video generation models. The main technical contributions, namely offline layer-wise sparsity profiling and online bidirectional co-clustering, are viewed as intuitive and useful, and the reported results show a consistently improved quality-efficiency trade-off over prior training-free baselines.

The main concerns focus on the strength of the theoretical justification, the precise characterization of prior work, and whether the benefit of the offline stage is always substantial in practice. These are reasonable concerns, and the paper would benefit from clearer positioning, improved presentation, and inclusion of several of the additional rebuttal experiments in the final version. That said, the rebuttal addressed most of the major questions by providing additional analyses on calibration sensitivity, ablations, overhead, memory, and comparisons to additional baselines, which substantially strengthens the paper. On balance, I find that the strengths outweigh the remaining weaknesses.